# Reservoir displacement by an invasive rodent reduces Lassa virus zoonotic spillover risk

Evan A. Eskew ![ORCID][1] ✉, Brian H. Bird[2], Bruno M. Ghersi ![ORCID][2,3], James Bangura[4], Andrew J. Basinski[1], Emmanuel Amara[4], Mohamed A. Bah[5], Marilyn C. Kanu[4], Osman T. Kanu[4], Edwin G. Lavalie[4], Victor Lungay[4], Willie Robert[4], Mohamed A. Vandi[6], Elisabeth Fichet-Calvet ![ORCID][7] & Scott L. Nuismer ![ORCID][8] ✉

The black rat (*Rattus rattus*) is a globally invasive species that has been widely introduced across Africa. Within its invasive range in West Africa, *R. rattus* may compete with the native rodent *Mastomys natalensis*, the primary reservoir host of Lassa virus, a zoonotic pathogen that kills thousands annually. Here, we use rodent trapping data from Sierra Leone and Guinea to show that *R. rattus* presence reduces *M. natalensis* density within the human dwellings where Lassa virus exposure is most likely to occur. Further, we integrate infection data from *M. natalensis* to demonstrate that Lassa virus zoonotic spillover risk is lower at sites with *R. rattus*. While non-native species can have numerous negative effects on ecosystems, our results suggest that *R. rattus* invasion has the indirect benefit of decreasing zoonotic spillover of an endemic pathogen, with important implications for invasive species control across West Africa.

Lassa fever, the human disease caused by Lassa virus, is endemic to sub-Saharan West Africa[1–3]. Estimates suggest that Lassa virus infects hundreds of thousands of people and causes thousands of deaths annually[1,4,5]. The Natal multimammate mouse (*Mastomys natalensis*) is the primary reservoir host of Lassa virus[1,6,7], although other rodent species may also play a role in viral maintenance[8–11]. Most human infections with Lassa virus are driven by rodent-to-human zoonotic spillover with little further transmission between people[3,12–15]. As such, understanding rodent ecology in West Africa, and in particular *M. natalensis*, is key to managing the threat of Lassa virus zoonotic spillover and improving public health in the region.

The black rat (*Rattus rattus*), a native of south Asia, is a notorious invasive species that is now distributed essentially worldwide. The precise timing of *R. rattus* introduction to West Africa is uncertain but may have occurred as early as the 15th century due to transcontinental maritime trade[16–18]. Regardless of the timing of introduction, black rats

were relatively common in major West African port cities, ranging from Senegal to Nigeria, by the early 20th century[19]. Following their arrival in coastal regions, *R. rattus* spread inland via both riverine and overland anthropogenic transportation networks[16,20–23]. As a result, the black rat was likely established in some inland areas of West Africa by the middle of the 20th century[20,22,24]. However, invasion across the region is still ongoing and may be facilitated by increasing availability of anthropogenic food subsidies and man-made structures that serve as rodent habitat[23,25].

The degree to which non-native *R. rattus* impact *M. natalensis* is an outstanding question with potentially significant implications for Lassa virus ecology and control. There are multiple reasons to think these two species interact. First, the two rodents currently co-occur at coarse geographic scales in West Africa, and they also share fine-scale habitat preferences: like *M. natalensis*[4,26,27], *R. rattus* frequently occupies human dwellings in its invasive African range[6,25,28–30]. Second, black rats

[1]Institute for Interdisciplinary Data Sciences, University of Idaho, Moscow, ID, USA. [2]One Health Institute, School of Veterinary Medicine, University of California - Davis, Davis, CA, USA. [3]Cummings School of Veterinary Medicine, Tufts University, North Grafton, MA, USA. [4]University of Makeni, Makeni, Sierra Leone. [5]Ministry of Agriculture and Forestry, Freetown, Sierra Leone. [6]Ministry of Health and Sanitation, Freetown, Sierra Leone. [7]Bernhard Nocht Institute for Tropical Medicine, Hamburg, Germany. [8]Department of Biological Sciences, University of Idaho, Moscow, ID, USA. ✉e-mail: eveskew@gmail.com; snuismer@uidaho.edu

are a detriment to native rodents in other invasion contexts. For example, black rats appear to depress native rodent densities in some habitats in Madagascar[31,32], and *R. rattus* has even been implicated in the complete extinction of island endemic *Rattus* species via disease-mediated competition[33]. Therefore, one argument holds that *M. natalensis* is negatively affected by competition with *R. rattus* and may in fact be completely displaced by it; such claims have dotted the literature on Lassa virus and African rodent ecology for decades[2,6,20,24,28]. This assumption of interspecific competition has even led some to suggest that purposeful introduction of *R. rattus* could serve as a form of biocontrol to manage Lassa fever[6,20]. Despite these strong assertions, there has been relatively little published evidence for antagonistic interactions between *R. rattus* and *M. natalensis*, and recent work that has started to fill this knowledge gap has not fully explored the consequences for zoonotic spillover of Lassa virus[27,28,34,35].

Here, we show how the ongoing *R. rattus* invasion of West Africa affects *M. natalensis* and, in turn, the potential for Lassa virus spillover to humans. We combine rodent trapping data from Sierra Leone and Guinea to examine associations between *R. rattus* and *M. natalensis*, quantifying these relationships using multiple data aggregation strategies (i.e., visit- and house-level analyses). In addition, we use Lassa virus infection data from *M. natalensis* to characterize zoonotic spillover risk at sites with and without *R. rattus*.

## Results

### Sites with *Rattus rattus* have fewer *Mastomys natalensis*

Across 28 study sites in Sierra Leone and Guinea, our primary rodent trapping dataset consisted of 678 *M. natalensis* captures and 140 *R. rattus* captures over 9588 trap-nights within houses. Site-level catch per trap ranged from 0 to 0.197 individuals per trap for *M. natalensis* (Fig. 1a) and from 0 to 0.235 individuals per trap for *R. rattus* (Fig. 1b). *M. natalensis* was rare at study sites < 100 km from the coast (Fig. 1a), and *R. rattus* was not detected at any of the five study sites > 200 km from the coast (Fig. 1b).

Sites with high *R. rattus* catch per trap tended to have low *M. natalensis* catch per trap (Fig. 1c). A visit-level Bayesian statistical

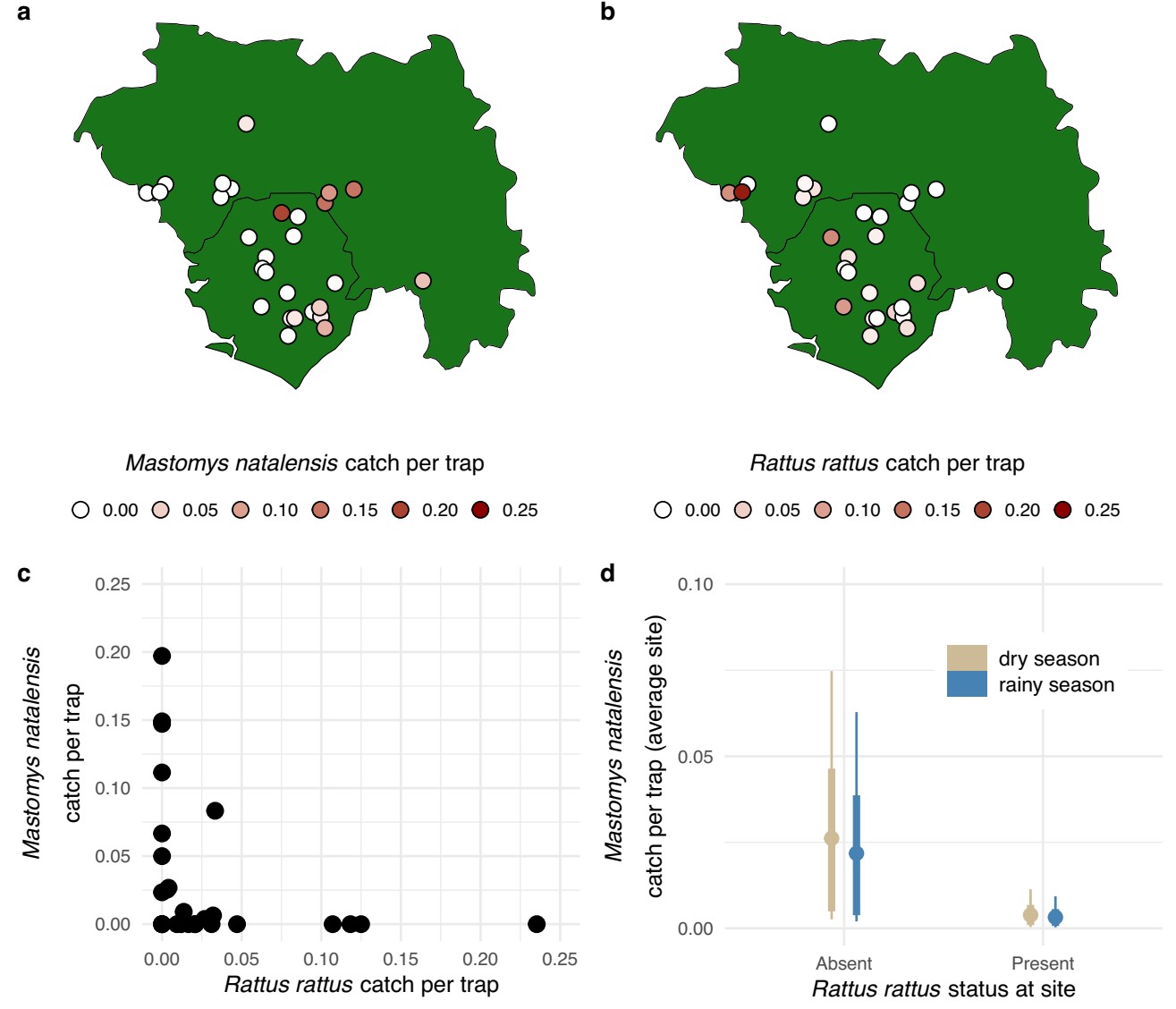

**Fig. 1 | Patterns of *Mastomys natalensis* and *Rattus rattus* catch per trap across 28 study sites in Sierra Leone and Guinea.** Map of catch per trap for *M. natalensis* (**a**) and *R. rattus* (**b**), and a scatterplot of the same data (**c**). Catch per trap was calculated using only house traps from a given site (*n* = 9588 trap-nights). **d** shows the implied values of *M. natalensis* catch per trap for sites without and with *R. rattus* present, as derived from a visit-level Bayesian statistical model (*n* = 20,000 posterior samples; see main text for details). Colors indicate sampling season, points indicate posterior means, thick lines represent 90% HPDIs, and thin lines represent 99% HPDIs. Source data are provided as a Source Data file.

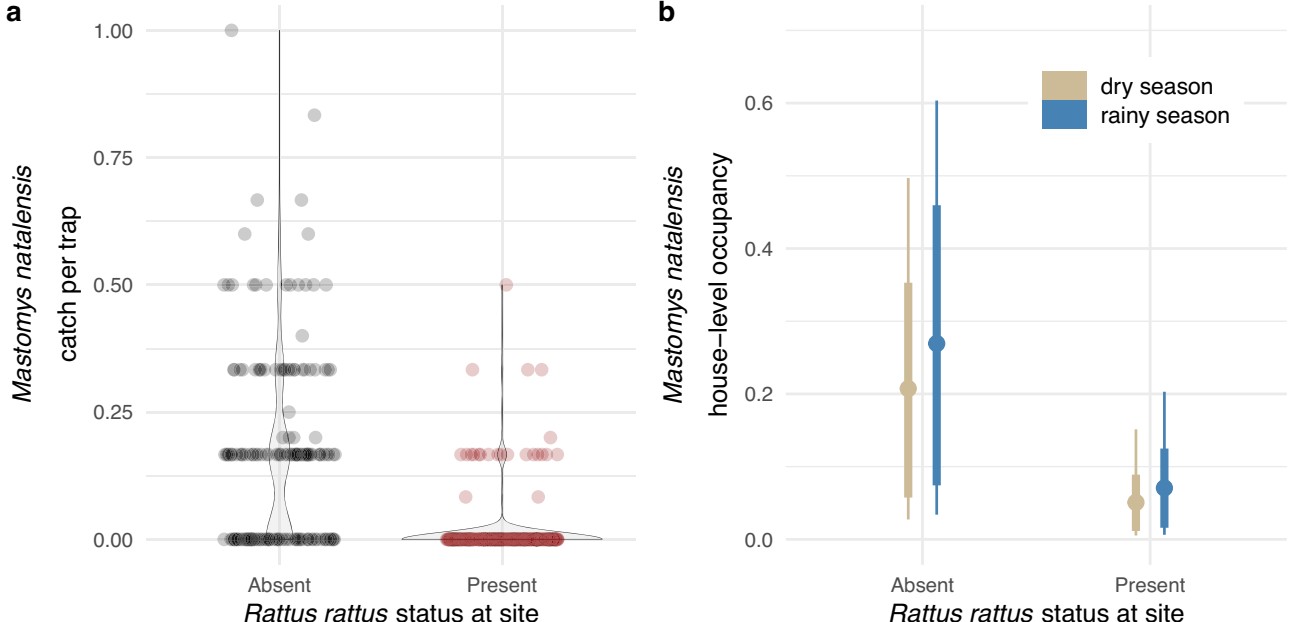

**Fig. 2 | House-level analyses suggest *Rattus rattus* negatively affects *Mastomys natalensis*.** Jitter plot showing *M. natalensis* catch per trap across 572 houses in Sierra Leone from sites where *R. rattus* is either apparently absent or known to occur (**a**). **b** shows posterior estimates for house-level *M. natalensis* occupancy probability for sites without and with *R. rattus* present (analysis based on 560 houses with repeated sampling suitable for occupancy modeling; $n = 100,000$ posterior samples). Colors indicate sampling season, points indicate posterior means, thick lines represent 90% HPDIs, and thin lines represent 99% HPDIs. Source data are provided as a Source Data file.

model accounting for seasonality supported the intuition that *R. rattus* presence at a site decreases *M. natalensis* catch per trap (coefficient for *R. rattus* presence effect = −1.90 [−3.47, −0.12], posterior mean [99% HPDI]; Fig. S1). This model also suggested that trapping in the rainy season may decrease *M. natalensis* catch per trap, but the magnitude of this seasonality effect was much smaller than the *R. rattus* effect and overlapped with 0 in the 99% HPDI (coefficient for rainy season sampling effect = −0.19 [−0.68, 0.31]; Fig. S1). Model-based estimates of dry season sampling implied an average site-level *M. natalensis* catch per trap of 0.026 [0.003, 0.075] when *R. rattus* was absent versus 0.004 [0.000, 0.011] when *R. rattus* was present (Fig. 1d).

The negative effect of *R. rattus* on *M. natalensis* was unique within the sampled rodent community. Estimates of the effect of four alternative rodent species on *M. natalensis* all overlapped with 0 in the 99% HPDI and even overlapped 0 in the narrower 80% HPDI (Fig. S2). As such, visit-level models did not suggest a strong directional influence of these rodent species on *M. natalensis* captures, as was the case for *R. rattus*.

Our findings on *R. rattus* were also robust with respect to the dataset chosen for analysis. We repeated the visit-level analysis considering all trapping effort at each site (i.e., traps set within homes and outdoors, representing 950 *M. natalensis* captures and 151 *R. rattus* captures over 36,759 trap-nights). With this broader dataset, the *R. rattus* effect on *M. natalensis* catch per trap was again strongly negative (−1.70 [−3.00, −0.44]; Figs. S3, S4) and larger in magnitude than the corresponding rainy season sampling effect, which overlapped with 0 in the 99% HPDI (0.16 [−0.50, 0.82]; Figs. S3, S4).

House-level analyses of trapping data from Sierra Leone provided additional support for a negative effect of *R. rattus* on *M. natalensis*. The raw data suggest that *M. natalensis* is less common in houses at sites where *R. rattus* is present: average house-level catch per trap of *M. natalensis* was 0.159 when *R. rattus* was absent versus 0.012 when *R. rattus* was present (Fig. 2a). A hierarchical Bayesian model captured a similar effect. In the model fit to house-level *M. natalensis* capture data, the estimated effect of *R. rattus* presence at a site was −1.18 [−2.91,

0.69] (Fig. S5). While this posterior overlaps with 0 in the 99% HPDI, 95.3% of this posterior's probability mass had support for negative values, indicating the influence of *R. rattus* presence on *M. natalensis* catch at the house level is likely negative. In this model, the rainy season sampling effect overlapped with 0 in the 99% HPDI and was smaller in magnitude than the *R. rattus* effect (coefficient for rainy season sampling effect = 0.30 [−0.54, 0.98]; Fig. S5). A supplementary house-level model constructed with a house-level *R. rattus* presence predictor gave qualitatively similar results with a negative, albeit slightly weaker, effect of *R. rattus* on *M. natalensis* (coefficient for *R. rattus* presence effect = −0.70 [−2.46, 0.84], 86.3% of posterior support for negative values; coefficient for rainy season sampling effect = 0.28 [−0.53, 0.94]; Fig. S7). When we repeated house-level analyses considering the presence of other rodent species at either the site (Fig. S6) or house (Fig. S8) levels as predictors, estimates of the effect of these alternative species on *M. natalensis* all overlapped 0 in the relatively narrow 80% HPDI. Thus, as with our visit-level analyses, these house-level results suggest that other rodent community members do not exert a consistent negative influence on *M. natalensis* in the same way as *R. rattus*.

An occupancy model fit to house-level *M. natalensis* detection data told a similar story. Our occupancy dataset consisted of trapping data from 560 houses in Sierra Leone, 183 from sites without *R. rattus* detected, of which 105 (57.4%) had *M. natalensis* detected, and 377 from sites with *R. rattus* detected, of which only 23 (6.1%) had *M. natalensis* detected. The detection parameters of our fit occupancy model implied that a single trap-night within a home had a 38% chance of detecting *M. natalensis*, conditional on the species being present (99% HPDI = [0.10, 0.70]). Further, our results indicated that increased trapping effort should generally increase *M. natalensis* detection (0.42 [−0.32, 1.15]). As such, deployment of three traps on a given night would be expected to boost mean detection over 50% (0.57 [0.44, 0.71]), while seven traps would be required to achieve a mean detection probability of at least 80% (0.83 [0.31, 1.00]), again conditional on *M. natalensis* being present. As expected, *M. natalensis* detection

**a**

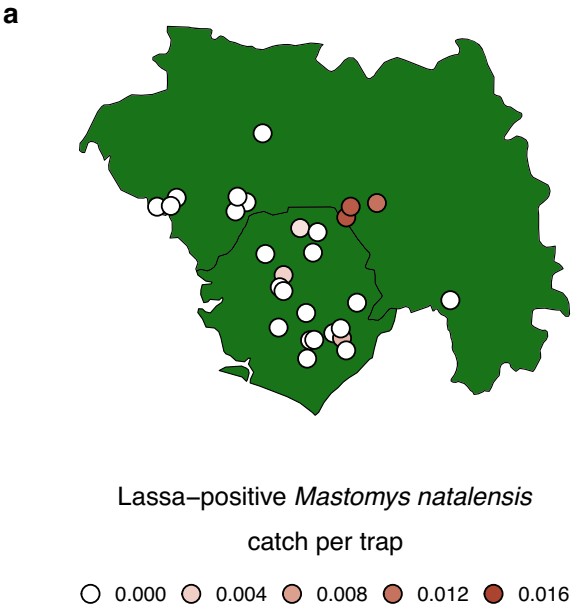

Lassa–positive *Mastomys natalensis*

catch per trap

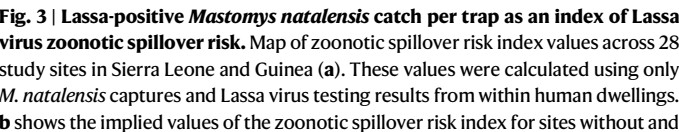

**b**

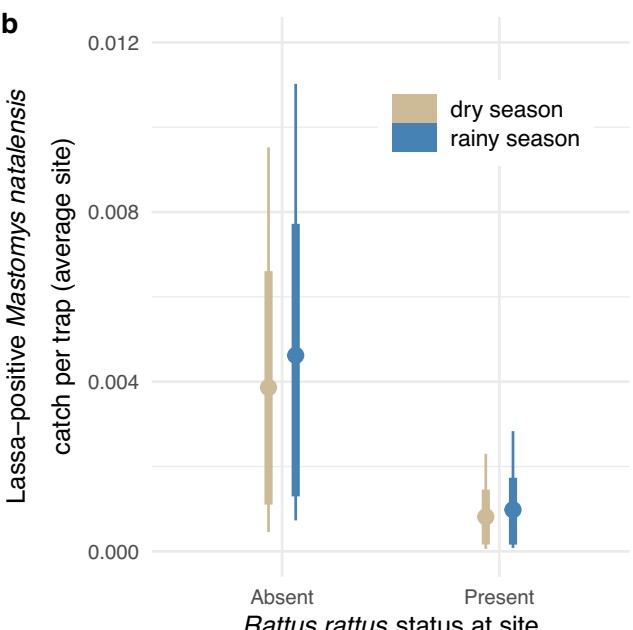

**Fig. 3 | Lassa-positive *Mastomys natalensis* catch per trap as an index of Lassa virus zoonotic spillover risk.** Map of zoonotic spillover risk index values across 28 study sites in Sierra Leone and Guinea (**a**). These values were calculated using only *M. natalensis* captures and Lassa virus testing results from within human dwellings. **b** shows the implied values of the zoonotic spillover risk index for sites without and with *R. rattus* present, as derived from a visit-level Bayesian statistical model (*n* = 20,000 posterior samples; see main text for details). Colors indicate sampling season, points indicate posterior means, thick lines represent 90% HPDIs, and thin lines represent 99% HPDIs. Source data are provided as a Source Data file.

decreased as progressively more *M. natalensis* were captured and removed from a given house (−1.10 [−1.77, −0.46]). In the occupancy portion of the model, the effect of *R. rattus* presence at the site had a negative influence on *M. natalensis* occupancy (−1.62 [−3.36, 0.12]). While this posterior overlaps with 0 in the 99% HPDI, 99.1% of the posterior's probability mass had support for negative values. The occupancy model implies that, in the dry season, a house at a typical site without *R. rattus* would have a 21% chance of being occupied by *M. natalensis* (0.21 [0.03, 0.50]) compared to a 5% chance of occupancy at a site with *R. rattus* (0.05 [0.01, 0.15]; Fig. 2b).

### Sites with *Rattus rattus* have reduced Lassa virus zoonotic spillover risk

Of the 639 house-captured *M. natalensis* that were tested for Lassa virus, 51 (8%) were Lassa-positive (had positive PCR test results that were confirmed by viral sequencing; Table S1). Site-level catch per trap of Lassa-positive *M. natalensis* within houses, which we treat as a proxy for Lassa virus spillover risk to humans, ranged from 0 to 0.015 across Sierra Leone and Guinea (Fig. 3a). A visit-level Bayesian statistical model revealed that this zoonotic spillover risk index was negatively related to the presence of *R. rattus* (−1.60 [−3.16, −0.15]; Fig. S9). Although not consistent across the entire 99% HPDI, catch per trap of Lassa-positive *M. natalensis* was positively related to rainy season sampling (0.19 [−0.68, 1.05]; Fig. S9), which contrasts with the negative relationship observed when analyzing *M. natalensis* catch within houses generally (Fig. 1d). Model results implied that the value of the zoonotic spillover risk index in the dry season at a typical site without *R. rattus* was 0.004 [0.000, 0.010] compared to 0.001 [0.000, 0.002] at a comparable site with *R. rattus* (Fig. 3b). No other rodent species had a similar influence on catch per trap of Lassa-positive *M. natalensis* (Fig. S10).

## Discussion

Research on invasive species and zoonotic disease generally focuses on the potential for introduced species to serve as hosts for endemic or novel pathogens, thereby increasing disease risk in invaded areas[36,37]. The same is true in the specific case of *R. rattus* invasion of Africa, where researchers have emphasized that the parasites hosted by black rats can pose a direct threat to human health[38-40]. Rarely have disease ecologists considered the complex ecological networks within which invasive species are situated and how their interactions within these networks might indirectly affect zoonotic disease risk[41,42]. Here, we demonstrate the negative effect of invasive *R. rattus* on native *M. natalensis* and link the invader's impact on the native rodent to reduced Lassa virus spillover risk for humans.

Our research adds to a body of work documenting *R. rattus* invasion of West Africa, which has likely been ongoing for centuries with substantial acceleration in recent decades. For example, by the early 20th century, black rats were already well-established in the major port cities of Dakar, Senegal and Lagos, Nigeria[19], two locations which span essentially the entire endemic range of Lassa fever. Given a foothold in coastal areas, *R. rattus* spread inland relatively rapidly: in Senegal, black rats occupied sites > 300 km from the coast by the mid-1980s[43]. Similarly, in our study region, *R. rattus* was present in inland portions of the eastern province of Sierra Leone in the early 1970s[6]. We documented *R. rattus* at 13 of 17 sites (76%) in Sierra Leone and at 5 of 11 sites (45%) in Guinea. Notably, we did not detect *R. rattus* at any of our five study sites that were > 200 km from the coast. However, we do note that *R. rattus* absence from sites in Guinea may not be reflective of the contemporary state of *R. rattus* invasion given that field data from Guinea were collected from 2002 to 2005. Indeed, more recent sampling from 2011 indicates the presence of *R. rattus* at two study sites in Upper Guinea where they were previously absent (pers. comm., E. Fichet-Calvet). Overall, these data suggest that *R. rattus* invasion is ongoing in this region of West Africa, particularly in northern and eastern Guinea where black rats may currently be absent or at low densities[28].

We provide quantitative evidence that the black rat invasion in Sierra Leone and Guinea has a negative impact on the primary host of Lassa virus, *M. natalensis*. Analyses of both visit- and house-level data

indicated that *R. rattus* presence decreases the number of *M. natalensis* captured. We found that *M. natalensis* is rare or absent in coastal regions of Sierra Leone and Guinea, in line with prior work on the topic[34]. Of the seven sampled sites that lie < 100 km from the coast, *M. natalensis* was only detected at two (29%), both in Sierra Leone. The absence of *M. natalensis* from western, coastal Guinea has been noted previously, and this gap in the distribution may be a natural range limit or a consequence of replacement by the congeneric species *M. erythroleucus*[7,29,30]. However, our data suggest this distribution pattern may also be consistent with exclusion by *R. rattus*[34], which is abundant at many of these same coastal sites.

In our models, we accounted for potential seasonal effects on *M. natalensis* captures, given that prior research has emphasized environmental influences on *M. natalensis* trap success and Lassa virus dynamics[5,28,44]. However, in our analyses, seasonality effects were relatively small, uncertain (all overlapped with 0 in the 99% HPDI), and sensitive to the particular dataset used for modeling. By contrast, the mean *R. rattus* presence effect was negative and larger in magnitude than the seasonality effect in all models we fit. Therefore, our results suggest that *R. rattus* presence may be more important in determining *M. natalensis* distribution and abundance than well-established environmental factors like seasonality.

Our use of an occupancy model to analyze house-level *M. natalensis* detection data afforded unique insights that are inaccessible without explicitly accounting for repeated sampling within sampling units (in this case, houses). Notably, the model-based estimate of mean *M. natalensis* detection probability for a single trap-night was 0.38. Consequently, modest trapping effort within homes is expected to give a relatively high likelihood of species detection, conditional on species presence (e.g., three traps for a single sampling visit gives a mean detection probability of 0.57). This information can be applied in field settings to guide rodent sampling, and encouragingly these results suggest that the majority of homes in our study were well-sampled. Of 572 homes in Sierra Leone included in this study, 482 (84%) were sampled using three traps over two consecutive nights, which implies a cumulative *M. natalensis* detection probability of ~0.82 for the two-night sampling scheme. Further, our model predicts that a substantial fraction of homes at sites without *R. rattus* are occupied by *M. natalensis*: the mean dry season estimate suggested 21% house occupancy, but the upper 90% HPDI ranged to 35% (Fig. 2b). In related work, 92.4% of survey respondents in the Bo District of Sierra Leone reported the presence of rodents in and around their home[25]. While these results are difficult to compare directly with our findings given that they represent individual-level reporting of rodent presence irrespective of species, it is clear that rodents, including *M. natalensis*, are extremely common in homes in the Lassa endemic zone, underscoring the significant potential for contact with humans.

Further research is needed to understand the exact mechanisms by which *R. rattus* excludes *M. natalensis* from otherwise favorable habitat. Thus far, targeted investigations of the two species have failed to settle on an explanation. For example, *M. natalensis* does not behaviorally avoid the scent of *R. rattus*[45], and, in Senegal, *M. natalensis* immunity measures do not show any differences between sites with and without *R. rattus*[46], suggesting the native species' physiology is not compromised when in the presence of the invader. As such, we are left to invoke general mechanisms that apply to *R. rattus* invasions more broadly, including various forms of competition and, potentially, predation of the native rodent species by black rats[47]. We also note the caveat that observed species' occurrence patterns can be shaped not only by biotic interactions, as we assume here, but also by the environment. In general, attempts to infer biological interactions from co-occurrence data alone are frustrated by numerous complicating factors including underlying environmental influences, unobserved interacting species, and the scale of sampling[48]. In our study system, where the spatial distribution of invading *R. rattus* may not yet be at equilibrium, there are likely to be multiple environmental variables that correlate with the occurrence of the two rodent species (e.g., elevation, rainfall). Therefore, while we cannot rule out the possibility that unmeasured environmental drivers affect the current distributions of *M. natalensis* and *R. rattus*, our study was meant to address a specific hypothesis that has a long history in the literature (i.e., that *R. rattus* negatively affects *M. natalensis*) using data that is richer and potentially more informative than strict presence/absence occurrence data (i.e., catch per trap of *M. natalensis*, a proxy for density).

Our modeling of Lassa virus zoonotic spillover risk suggests that sites where *R. rattus* is present tend to have lower spillover pressure, consistent with the observation that *M. natalensis*, which most directly drives zoonotic spillover in this system, is depressed by *R. rattus*. These findings were motivated by and agree with prior work showing that villages with a greater proportion of *M. natalensis* in the local rodent community tend to have the highest Lassa seroprevalence rates in humans[34]. While local people in invaded regions may seek to eliminate *R. rattus* populations because of their activity as household pests[25], our results show that black rats actually seem to reduce human exposure to Lassa virus, a counterintuitive ecosystem service provided by a non-native species[49–51]. Nonetheless, given the potential for *R. rattus* to harm human and ecosystem health in other ways, we stress that we do not advocate for biocontrol strategies that would attempt to use this species to manage zoonotic spillover of Lassa virus. For example, it is critical to recognize that our findings of reduced spillover at sites with *R. rattus* are specific to the Lassa fever system and that invasive *Rattus* do host other zoonotic pathogens that can threaten humans[16,52]. Further, the negative impacts of invasive *R. rattus* on native ecosystems, for example through predation and competition, are widespread and well-documented[47,53].

We also highlight several caveats that may complicate the relationship we suggest between *R. rattus* and Lassa virus zoonotic spillover risk. First, *R. rattus* may not completely displace *M. natalensis* from village sites, at least on short time scales. For example, in our study we found some coexistence between *M. natalensis* and *R. rattus*, with 12 of 28 study sites having both species present, a situation that has also been observed at other West African locations[35,43,54]. In these ecological contexts, *M. natalensis* may be less abundant, particularly in human dwellings, due to the presence of *R. rattus*, but it would still represent a zoonotic disease threat. In such areas, people may be exposed to Lassa virus later in life as a consequence of reduced *M. natalensis* abundance or activity in households. Consistent with this idea, coastal sites in Guinea (where *R. rattus* is common and *M. natalensis* is rare) show positive relationships between age and Lassa seroprevalence in humans, whereas inland sites (where *R. rattus* is rare and *M. natalensis* is common) show universally high seroprevalence, including in the youngest individuals sampled[55]. Intriguingly, if Lassa virus infection later in life leads to more severe disease outcomes, a reduction in spillover pressure could actually increase overall disease burden within human populations. Second, even in scenarios where *R. rattus* does completely displace *M. natalensis*, there is the potential for other native rodent species to host and potentially maintain the virus in the local community[8,10], compensating for the absence of *M. natalensis*. Our zoonotic spillover risk index focuses exclusively on *M. natalensis* as the primary host of Lassa virus, but viral testing in the rodent community more broadly would provide the most comprehensive picture of zoonotic risk in areas with and without *R. rattus*. Finally, we note that our spillover index is our best attempt to characterize Lassa virus zoonotic spillover risk, but it does not directly incorporate information on viral exposure in the human population. Future work could build on our results by evaluating Lassa virus seroprevalence in humans and correlating this measure with local *R. rattus* population characteristics[34].

In sum, our results suggest that Lassa virus ecology cannot be understood without consideration of the entire rodent community in

the Lassa endemic region. In addition to the fact that rodent species aside from *M. natalensis* may carry Lassa virus[8,10], the presence of specific species, such as invasive *R. rattus*, may indirectly modify Lassa virus dynamics via effects on *M. natalensis*. Future work on rodents and Lassa virus should aim to census the entire rodent community and pay special attention to a region's invasion status, noting that *R. rattus* may not be the only invasive rodent of importance[27,34,35,43]. Further, disease interventions in the Lassa system should be evaluated with consideration of potential unintended outcomes generated by interactions between *M. natalensis* and *R. rattus*. For example, while rodent control efforts in West Africa aim to bolster food security and public health[25,56], any methods that disproportionately harm *R. rattus* may indirectly benefit *M. natalensis*, thereby increasing Lassa virus zoonotic spillover risk[28]. Ultimately, biological invasions must be considered alongside climate and land use[57–59] as one of the major global change factors shaping Lassa distribution now and in the future.

## Methods

### Data overview

We combined rodent trapping data from Sierra Leone (*n* = 17 sites) and Guinea (*n* = 11 sites), two countries where per-capita risk of Lassa virus infection in humans is thought to be particularly high[5]. Sites represent distinct villages where extensive rodent trapping occurred both within human dwellings and in the surrounding landscape over multiple site visits and trapping nights. Sites were visited one to six times, and total site-level trapping effort ranged from 211 to 6303 trap-nights. The dataset from Sierra Leone consisted of trapping results from traps placed in one of three locations: inside homes, along transects outside of homes but within the village perimeter, or along transects outside of the village perimeter. Trapping effort was adjusted based on village size to avoid over- or under-sampling. During each site visit, traps were set at dusk and checked during the early morning, generally for two consecutive nights. Traps were placed inside houses (three traps per house), along transects within the village (transect size dependent upon village size), and in the surrounding field/bush (transect size same as in the village). Trapping in Sierra Leone occurred between July 2019 and February 2021. Animal work in Sierra Leone was reviewed and approved by the University of California, Davis Institutional Animal Care and Use Committee (IACUC; protocol no. 22696) and was conducted in collaboration with the Sierra Leone Ministry of Health and Sanitation and the Ministry of Agriculture and Forestry under permit no. CONF/LSD/02/17. Prior to rodent sampling, project team members engaged communities to ensure community awareness and approval of the proposed work via community leader-led workshops and meetings involving government and traditional leaders, community liaison officers, and district-level Ministry of Health and Sanitation and Ministry of Agriculture and Forestry representatives. No work was conducted until local community approvals were obtained. Rodent sampling in the peridomestic setting and within human households was only conducted after verbal permission was obtained from each household owner. Sampling that generated the Guinea dataset was broadly similar, except that trapping effort was targeted within homes, in cultivated areas, and in forests (see more detail in refs. [30,44,60]). Trapping took place in Guinea between October 2002 and February 2005. Rodent trapping in Guinea was authorized by the Ministry of Public Health (permit no. 2003/PFHG/05/GUI). Prior to data collection at a given site, the project was presented to the community, including the different phases and components. During the presentation, we confirmed the agreement of the village chief and the community to allow rodent trapping. Finally, traps were only placed in a house if permission of the individual house owner was obtained.

Following rodent capture, animal processing was necessary for species identification and the collection of tissue samples for later viral testing (see "Consequences for Lassa virus zoonotic spillover risk" section below). In Sierra Leone, captured animals were transported to a designated area outside of the village, and, using personal protective equipment and operating procedures consistent with BSL-3 biosafety practices, animals were anesthetized with isoflurane or halothane. All animals trapped during the first two rounds of field work and those captured inside houses were euthanized with an overdose of iso-flurane/halothane, and specimens and tissues (oral/urogenital/rectal swabs, blood, lung, liver, and spleen) were collected, stored in liquid nitrogen, and transported to the University of Makeni for pathogen testing. Other animals were anesthetized and only non-invasive samples were collected (oral/urogenital/rectal swabs and blood). Species identification in the field was based on morphological characteristics, and identifications were confirmed by cytochrome *b* sequencing. Similarly, in Guinea, animals were euthanized using halothane and then necropsied outside of villages in dedicated spaces using field procedures analogous to BSL-3 methods[61,62]. Animals were initially described morphologically and subsequently identified via a PCR assay targeting cytochrome *b*[63]. Blood was collected via cardiac puncture, stored in liquid nitrogen, and sent to the University of Marburg for further testing (see detailed procedures in refs. [7,44,64]).

### Associations between *Rattus rattus* and *Mastomys natalensis*

We explored associations between *R. rattus* and *M. natalensis* by analyzing trapping rate data as a proxy for rodent density. Recognizing that the study designs in Sierra Leone and Guinea emphasized different trapping locations outside of homes, we chose to focus on trapping data from houses for the majority of our statistical analyses. House trapping methods were directly comparable between countries, and *M. natalensis* abundance within homes is arguably more relevant to Lassa virus zoonotic spillover than *M. natalensis* abundance at sites generally, given the importance of rodent-human contact within homes[3,25,62]. We conducted statistical analyses at two complementary scales, the visit-level and the house-level, relying on slightly different datasets and methods at each scale.

**Visit-level analyses.** To analyze house trapping data across the 28 sites in Sierra Leone and Guinea, we conceptualized rodent sampling as fundamentally a Poisson process, with each trap representing an exposure capable of recording some number of *M. natalensis* captures. In other words, the result of each trap-night (exposure) can be expressed as a non-negative, integer count of *M. natalensis* captured. Consequently, we modeled these data using a hierarchical Bayesian model with a Poisson outcome and a log link function (Figs. S11, S12). While our primary aim was to understand the effect of *R. rattus* on *M. natalensis* captures, we also wanted to account for potential seasonal effects on *M. natalensis* abundance and activity that could influence capture success[28,44]. As such, we first organized the trapping data at the visit-level, with each observation representing trapping results from a given site visit (*n* = 71 unique site visits that involved house trapping) and each visit coded as having occurred in the dry season (December-April) or rainy season (May–November). Our dry/rainy season definition closely aligns with prior literature[28,44,56], and we additionally confirmed using remotely-sensed data that precipitation[65] and temperature[66] differed significantly between dry and rainy season site visits using this definition of seasonality (Fig. S13). Some site visits did not result in *R. rattus* capture even when *R. rattus* was known to occur at the site based on other visits. Therefore, in our models, we opted to formulate the *R. rattus* effect as a site-level binary variable that indicated whether *R. rattus* had been captured at the site during any site visit. In full, our visit-level model included an intercept term, a main effect of *R. rattus* (formulated as a binary, presence/absence variable indicating whether or not *R. rattus* was ever captured at that site), a main effect of season (formulated as a binary variable indicating whether or not the site visit occurred in the rainy season), and an offset term of total trap-nights of the site visit to properly scale the outcome data (Fig. S11). In addition, we included varying effects (i.e., random

effects) of site and visit in this model (Fig. S11). Given that trapping effort was unbalanced across sites and visits, varying intercepts allowed us to account for this variation, in effect recovering inference about a typical site in the model's intercept term[67].

For our visit-level Bayesian model, we sought to construct an informed prior for *M. natalensis* catch per trap[68], given a priori knowledge that *M. natalensis* capture events are typically rare (i.e,. far less than one individual per trap-night). To do so, we reviewed the literature and collated 158 study- or site-level estimates of *M. natalensis* catch per trap. These estimates came from 10 independent studies and spanned rodent trapping efforts across eight African nations. The median of these *M. natalensis* catch per trap estimates was 0.044, while the mean was 0.077 (range 0−1). Therefore, for the model intercept parameter's prior distribution, we chose a normal distribution with a mean of −3.1 and a standard deviation of 1.1 (Figs. S11, S12). Accounting for the model's log link function, this implies a prior distribution for the Poisson rate ($\lambda$) with a median of 0.045, a mean of 0.082, and essentially all probability mass (-0.998) below 1 (Fig. S12). As such, this prior closely reflects preexisting evidence regarding *M. natalensis* catch per trap. For the priors for the main effects of *R. rattus* presence and rainy season sampling, we chose normal distributions with means of 0 and standard deviations of 1, placing equal prior weight on a positive or negative influence of these predictors on *M. natalensis* capture (Fig. S11). Finally, we modeled the site- and visit-level varying intercepts from normal distributions with a mean of 0 and standard deviation hyperparameters estimated using exponential distributions with a rate equal to 1 (Fig. S11)[67].

We fit multiple alternative visit-level models as robustness checks of our initial modeling results. First, we investigated whether rodent species other than *R. rattus* might influence capture success of *M. natalensis*. Thus, we constructed alternative visit-level models where the *R. rattus* presence/absence predictor was replaced with the presence/absence of another common rodent species. Specifically, we tested models where each rodent species with more than 100 total captures in the dataset was used as a presence/absence predictor. Focusing on only these commonly captured rodents resulted in consideration of four additional visit-level models with predictors representing the presence/absence of *Mastomys erythroleucus* ($n = 377$ captures), *Mus mattheyi* ($n = 135$ captures), *Praomys daltoni* ($n = 114$ captures), and *Praomys rostratus* ($n = 167$ captures). Second, we fit the visit-level model using all trapping data available from the 28 study sites. Although our primary analyses excluded traps placed outdoors because sampling strategies were different in Sierra Leone and Guinea, we performed this additional analysis that included all trapping data to ensure our results remained robust.

We fit Bayesian models with Stan[69], using the 'cmdstanr' interface in R[70]. We ran four independent Markov chains for each model, and each chain was sampled for 5000 post-warmup iterations. As a result, our inferences draw on a total of 20,000 posterior samples per model. We verified model convergence using a split-$\hat{R}$ statistic, which compares between-chain parameter variance to within-chain variance, confirming all parameters had a value approaching 1[71]. In addition, we plotted parameter traces and confirmed lack of divergent transitions during model fitting. We summarize model parameters using 99% highest posterior density intervals (HPDIs)[67], communicating nearly the full width of the posterior distribution. Further, in some cases we directly report the proportion of posterior probability mass that lies within a key range (e.g., the proportion of probability mass < 0, indicating a negative influence of a predictor variable on the outcome).

**House-level analyses.** To further investigate the effect of *R. rattus* on *M. natalensis*, we performed a more granular analysis at the level of individual houses. Since trapping information at the scale of individual houses was only available from Sierra Leone, our house-level analyses were restricted to the 17 sites from that country.

Our first approach to analyzing the house-level data was directly analogous to our visit-level analyses. Specifically, we implemented a hierarchical Bayesian model identical to our visit-level Bayesian model but fit to house-level data and with the addition of house-level varying effects. The house-level varying effects used the same priors as the site- and visit-level varying effects. Importantly, this model treats a house as exposed to *R. rattus* if *R. rattus* was known to be present at the house's site, regardless of the specific locations where *R. rattus* were captured within that site. To validate the model results obtained using these assumptions, we also constructed a supplementary house-level model that instead used a house-level *R. rattus* presence predictor. In this model, houses were only coded as exposed to *R. rattus* if *R. rattus* was detected at that specific house during the study period. As with visit-level models, we checked the robustness of the results from these two house-level models by fitting alternative models that included either site-level or house-level predictors corresponding to other common rodent species in the sampled community. Because *Mus mattheyi* was never captured within houses in Sierra Leone, these supplemental house-level analyses only considered *Mastomys erythroleucus*, *Praomys daltoni*, and *Praomys rostratus* as alternative rodent species. All house-level models were fit using procedures identical to the other Stan models as described above.

The particular structure of the house-level data offered further avenues for analysis. More specifically, houses in Sierra Leone were sampled over multiple nights, with the most frequent sampling scheme placing three traps in a house over two consecutive nights. As such, these house-level data were amenable to a single-species occupancy analysis focused on the occupancy status of *M. natalensis*. In this analysis, each individual house represents a distinct "site" whose latent occupancy status of *M. natalensis* (present/absent) is estimated while accounting for imperfect detection. The outcome data for this analysis was an *M. natalensis* detection matrix (1/0; detected/not detected) with rows equal to the total number of houses ($n = 560$) and columns equal to the maximum number of consecutive trapping nights at a house ($n = 4$). Occupancy models allow distinct predictor sets to be used for the occupancy and detection components of the model. For the occupancy component of the model (i.e., those factors predicting whether or not *M. natalensis* actually occupies a given house), we used an intercept term, a main effect of *R. rattus* presence/absence at the site (village), and a main effect of sampling season (rainy vs. dry season). In addition, we included site-level (village-level) and visit-level varying intercepts structures. In essence, the predictor structure for the occupancy component of the model mirrors the predictor structure for the hierarchical Bayesian models described above.

For the detection component of the model, we wanted to account for two important factors that we expected to influence the probability of detecting *M. natalensis* within a house: nightly trapping effort and cumulative *M. natalensis* caught in the house. In an occupancy model, the dimensions of the detection covariate matrices matches that of the outcome data: in our context, there are distinct detection covariate values for each house by trapping night combination. Given that placing more traps in a house on a given night should increase the chances of detecting *M. natalensis* (conditional on *M. natalensis* presence), we incorporated nightly trap count values within each house as a detection covariate. In addition, an underlying assumption of occupancy models is that sites are closed such that their occupancy status does not change throughout the sampling period. In our case, this would translate to individual houses either being occupied or unoccupied by *M. natalensis* throughout the sampling duration. This assumption may not hold if successful trapping efforts result in the removal of animals, as they did in our sampling context. While we cannot easily account for occupancy status changes at individual houses over time, we can use cumulative *M. natalensis* catch at each house on each trapping night as a detection covariate to model detection bias. Including this detection covariate allowed us to

explicitly model the possibility that detection of *M. natalensis* decreases as increasing numbers of *M. natalensis* are captured and removed from a home.

To fit the house-level occupancy model, we used the R package 'spOccupancy'[72]. This package uses a Bayesian framework but differs from Stan in that it implements a version of Gibbs sampling customized for fitting occupancy models. For all occupancy and detection parameters, we used a normal distribution with a mean of 0 and a standard deviation of 1 as the prior with the exception of the varying effects structures, which 'spOccupancy' models using an inverse-Gamma distribution (in our case, with the shape parameter set to 3 and the scale parameter set to 1). We fit the occupancy model using four Markov chains with 40,000 samples each (15,000 discarded as burn-in), for a total of 100,000 posterior samples. As with Bayesian models fit using Stan, we used the $\hat{R}$ statistic and parameter trace plots to assess model convergence.

## Consequences for Lassa virus zoonotic spillover risk

Finally, we sought to connect rodent population status to human risk for infectious disease. More specifically, we aimed to quantify Lassa virus zoonotic spillover pressure at sites across Sierra Leone and Guinea. We reasoned that if *R. rattus* presence negatively affects *M. natalensis*, this would have knock-on impacts on human risk for Lassa fever, given that *M. natalensis* is the primary reservoir for Lassa virus. Lacking data on Lassa seroprevalence within human populations at our study sites, we instead focused on generating a proxy index for Lassa virus zoonotic spillover risk[5]. Initially, we conceptualized a zoonotic spillover index consisting of *M. natalensis* density (catch per trap) multiplied by the Lassa virus prevalence within *M. natalensis*. However, such a measure can be simplified to the catch per trap of Lassa-positive *M. natalensis* (total *M. natalensis* captures, the numerator of *M. natalensis* catch per trap and the denominator of Lassa virus prevalence, cancels out in the calculation). Therefore, we used the catch per trap of Lassa-positive *M. natalensis* as our index of Lassa virus zoonotic spillover risk. Our index assumes *M. natalensis* is the only rodent in the community contributing to zoonotic Lassa virus transmission, an assumption that seems reasonable given *M. natalensis* is widely considered the primary reservoir host of the virus[1,6,7]. While studies have occasionally presented evidence of Lassa virus infection in rodents identified as *R. rattus*[73], these have been interpreted as rare, transient cross-species transmission events[28], and researchers generally report a lack of infection in this species, particularly in our study region of Sierra Leone and Guinea[6,7,54].

Further, we acknowledge that zoonotic spillover risk must be operationalized carefully as the precise meaning of the term "risk" varies across disease ecology contexts. We find a useful definition in the terminology of ref. 74 who conceptualize disease risk as being driven by a disease hazard (potential source of harm), conditional on exposure to that hazard. Under this framework, our zoonotic spillover risk index might be considered a measure of disease hazard because it does not explicitly account for human exposure to the harm that we quantify (*M. natalensis* infected with Lassa virus). However, we emphasize that our zoonotic spillover risk analysis only includes rodent trapping data gathered from within human habitations. Thus, our zoonotic spillover risk index quantifies disease hazard within the specific context (the domestic setting) where most humans are presumably exposed to Lassa virus. We therefore argue that our measure conditions on pathogen exposure to deliver a reasonable approximation of Lassa virus zoonotic spillover risk.

Our Lassa virus zoonotic spillover risk analysis relied on Lassa virus infection data from sampled *M. natalensis*. For rodents sampled in both Sierra Leone and Guinea, Lassa status was initially determined using PCR. In Sierra Leone, oral/urogenital/rectal swabs and blood samples were all tested using the real-time PCR primers and protocol of ref. 75. If any specimen from an individual rodent returned positive

results, that animal was tentatively considered Lassa virus positive. In Guinea, rodent blood was tested for Lassa virus using conventional RT-PCR according to the protocol of ref. 76. To ensure the most accurate Lassa virus infection data, we only treated an *M. natalensis* individual as Lassa-positive if a positive PCR test result was further supported by the recovery of Lassa virus sequence from the same individual. We relied on viral sequencing data generated in prior work for this confirmation: PCR-positive animals from Sierra Leone were confirmed by the generation of Lassa L and S segment sequences[77,78], while PCR-positive animals from Guinea were confirmed by the generation of Lassa nucleoprotein (NP) sequence[79] (Table S1). The vast majority of all *M. natalensis* captured within homes were tested for Lassa virus (639 of 678 animals; 94%). Full testing results were unavailable for *M. natalensis* from two sites, both of which had no *R. rattus* detected. Our index therefore is likely to slightly underestimate Lassa virus zoonotic spillover risk for sites without *R. rattus*.

In sum, our Lassa virus zoonotic spillover risk model is identical to our visit-level model including only house trap data with one key difference: the zoonotic spillover risk model uses Lassa-positive *M. natalensis* count as the outcome as opposed to total *M. natalensis* count. As with the visit-level model, the zoonotic spillover risk model included binary main effects predictors of *R. rattus* presence at a site and sampling season as well as site- and visit-level varying intercepts. Priors were identical to the visit-level model, we fit supplementary zoonotic spillover risk models using alternative rodent presence predictors as with the visit-level model, and we fit the zoonotic spillover risk models using the previously described Stan methods.

## Reporting summary

Further information on research design is available in the Nature Portfolio Reporting Summary linked to this article.

## Data availability

All data used to generate the analytical results in this manuscript are openly available via the project GitHub (https://github.com/eveskew/rat_invasion) and Zenodo (https://doi.org/10.5281/zenodo.10946459) repositories. GenBank accession codes for Lassa virus sequence data from infected *Mastomys natalensis* are provied in Table S1. CHIRPS precipitation (https://www.chc.ucsb.edu/data/chirps) and MODIS land surface temperature (https://lpdaac.usgs.gov/products/mod11c3v061/) data used to validate our seasonality variable are freely available online and are also included in our project's Zenodo repository (https://doi.org/10.5281/zenodo.10946459). Source data are provided with this paper.

## Code availability

Code supporting this manuscript is openly available via GitHub (https://github.com/eveskew/rat_invasion) and Zenodo (https://doi.org/10.5281/zenodo.10946459).

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

## Acknowledgements

This work was supported by the National Institutes of Health through grant number R01GM122079 (S.L.N.), the National Science Foundation's Division of Environmental Biology through grant number 2028162 (S.L.N.), the Defense Advanced Research Projects Agency through grant number D18AC00028 (B.H.B., S.L.N.), and the European Union through INCO-DEV grant number ICA4-CT2002-10050 (E.F.-C.). The funders had no role in study design, data collection and analysis, the decision to publish, or preparation of the manuscript.

## Author contributions

E.A.E., A.J.B. and S.L.N. conceived of the analyses. B.H.B., B.M.G. and J.B. funded and guided data collection in Sierra Leone. E.F.-C. funded and led data collection in Guinea. E.A., M.A.B., M.C.K., O.T.K., E.G.L., V.L., W.R. and M.A.V. facilitated and conducted fieldwork in Sierra Leone. A.J.B. contributed to data processing and cleaning. E.A.E. led analyses and drafted the manuscript. All authors helped craft the final version of the analyses and manuscript.

## Competing interests

The authors declare no competing interests.
