## [Peer Review File · Nature Communications]

Reservoir displacement by an invasive rodent reduces Lassa virus zoonotic spillover riskReviewers' comments:

Reviewer #1 (Remarks to the Author):

This manuscript focuses on the impact of non-native rodent species (*Rattus rattus*) on the ecology of native *Mastomys N.*, and how this affect spillover of Lassa fever. I read the manuscript with interest; Assessing ecological changes in the reservoir is extremely important and the results are certainly intriguing. I think the authors did a good job in terms of fieldwork and analysis, but I believe that they need to improve the text, especially in the methodology session.

Below are my comments:

- Methods: I would really appreciate if the authors can explicitly write down the equations of the Poisson model. I know that many researchers just explain the model in a narrative way rather than mathematical equations, but I don't think that this is a good practice. It makes the methodology less transparent and hard to follow. Please write down your equations and explicitly relate your assumptions with the equations. You can do this in the supplementary material.
- I presume that both *Mastomys N.* and *Rattus Rattus* exhibit seasonal patterns. If so, are the pattern of seasonality the same for both species? Shouldn't these be included in the analysis? Please explain.
- Line 88-90, when the author say Sampling that generated the Guinea dataset was generally similar, except trapping effort was targeted within homes, in cultivated areas, and in forests. Please explain why the sampling in Guinea was done slightly different.
- Line 111-113, See point above. Explicitly write down in mathematical terms the rate of the Poisson process, intercept and offset terms along with their interpretation.
- Line 111-113, please explain the rationale to model the effect of *Rattus Rattus* in terms of presence/absence alone and not their abundance (I guess you can catch 1-2 *Rattus Rattus* at most, so your choice is reasonable to me, but I think you need to explain this).
- Line 122-124. If the mean was 0.077, I am not sure if a median of 0.044 is compatible with a Poisson distribution (there are some approximations in the literature). Have the authors checked that the empirical data qualitatively follow a Poisson distribution?
- Line 128 and line 131. I can see why mean of zero, but why standard deviation of 1?
- Line 136-144 and 188-195. I believe it is standard practice to show some diagnostic of the MCMC, you can do this in the supplementary materials.
- Line 162-171. I must admit that I am not clear what the authors exactly did. For example:
 - when they say rows equal to the total number of houses, do they refer to total number of houses for different sites or what?
 - I am not clear if they did a spatial analysis, if so, do they take into account spatial clustering?
 - Do they treat each occupancy as Bernoulli process?
 - Is this single species or multispecies?
 - What are the predictors? (I guess the nightly trapping effort and cumulative *M. natalensis* caught in the house but not clear).
- Line 280-281. When they say the estimated effect of *R. rattus* presence at a site was -1.12 [-2.90, 0.74]. I am not clear what the numbers exactly measure (although the interpretation is ok). Same when they say and posterior overlaps with 0 in the 99% HPDI, posterior of what? Again, writing equations would massively help.

Reviewer #2 (Remarks to the Author):

This paper provides a contemporary analysis of a mix of self and published data on rodent trapping in Sierra Leone and Guinea in the context of LFV spill over risk - focused on the role of invading *R. rattus*. The authors I believe correctly identify the need for community rodent ecology research in this area and the importance of understanding shifting habitat and population dynamics across LFV infected landscapes, villages and towns. They attempt to use available data and some self collected material to prove the point but I fear this is a step too far. The use of fancy statistical methods and modelling will not solve this. More systematic trapping methods are the only way to get to the truth and across the whole ecological and epidemiological landscape for LFV. My recommendation is to reduce the scope of what the paper is attempting to do and focus where the comparisons between own and other data are valid. I see this probably only in the trapping in housing and view on *M. natalensis* and *R. rattus* in this context. There is no crime in speculation in discussion on the core subject but to make the sort of statements suggesting that invasion of *R. rattus* might be good outcome is surprising given the isolated issue examined. I currently cannot accept this publication with extensive revision although I see a lot of merit in it so would encourage a more careful scope of work. I have included some specific comments on the PDF for your consideration.

Reviewer #3 (Remarks to the Author):

Summary

In this work, the authors use rodent trapping data from 28 sites in Sierra Leone and Guinea to evaluate the impact of the presence of the invasive species *Rattus rattus* on the density of *Mastomys natalensis*, a rodent species considered the primary host of Lassa virus. By building hierarchical models based on house-level and site-level data, as well as an occupancy model based on house-level data. They identify a negative effect of *R. rattus* presence on (1) *M. natalensis* catch per trap numbers both at site- and house-levels (hierarchical models), (2) *M. natalensis* probability of detection (occupancy model), (3) number of catch per trap of Lassa-positive *M. natalensis* -used here as an indicator of Lassa virus spill-over risk.

General Comment

This interesting work adds to previous studies that have started to describe the impact of *R. rattus* presence on *M. natalensis* in Guinea, Benin, and Niger, and provides a first assessment of a potential effect on Lassa spill-over risk. There are, however, several limitations, both in the analyses and conclusions, that are of concern at this stage.

Major Comments

Results Section 1: Contemporary sites with *Rattus rattus* have fewer *Mastomys natalensis*

-Confounding variables:

In their hierarchical model, the authors account for the effect of several variables on catch per trap numbers including *R. rattus* presence, total trap-nights number, heterogeneity of sampling efforts across sites. However, Several other factors (environmental, temporal) are known to affect *M. natalensis* populations and in particular, seasonal variations in population numbers and localisation (e.g. in [1]). I would recommend accounting at least for this possible confounding variable in the analysis.

-Specificity of the effect:

The main point of this study is to demonstrate a specific (and negative) effect of *R. rattus* on *M. natalensis* presence. While the authors do identify an effect of *R. rattus* on *M. natalensis*, it is not clear that this effect is specific to *R. rattus*. Performing similar comparisons using a "control" population (another rodent species present at study sites) would demonstrate the specificity of the

R.rattus/M.natalensis antagonism.

-Impact of the prior:

The authors describe using an informed prior that reflects pre-existing evidence regarding M. natalensis catch per trap. Did the authors evaluate how sensitive their estimates were to this choice of prior?

Results Section 2: Comparison of contemporary and historical sampling for Mastomys natalensis

In this section, the authors state that they "recovered some evidence for temporal declines in M. natalensis density in Sierra Leone's eastern province" however, model estimates of average site-level M. natalensis catch per trap are close with broadly overlapping ranges (0.033 [0.019, 0.054] historically - 0.024 [0.007, 0.050] currently) and an 99% HPDI for the effect of contemporary sampling that overlaps with 0. Given the weakness of the evidence, I would recommend a more cautious formulation of the results.

Results Section 3: Contemporary sites with Rattus rattus have reduced Lassa virus spillover risk

As for section 1, this part of the analysis would be much stronger if accounter for seasonal variations and assessing the specificity of the effect of R. rattus.

Minor comments

l43: "while invasion is ongoing in other locations [20]" Can you specify which other locations ? In the current form, it is unclear which areas are the ones where invasion is ongoing (and which areas are the ones where R. rattus is established)

l44: "a outstanding" perhaps the authors rather meant "an outstanding"

l116-117: "in effect recovering inference about an typical site in the model's intercept term" this sentence seems to need editing

l392-393: "our results show that black rats may actually protect against Lassa exposure" I would recommend using "reduce" rather than protect against

References

[1] Fichet-Calvet E, Lecompte E, Koivogui L, Soropogui B, Dore A, Kourouma F, et al. Fluctuation of abundance and Lassa virus prevalence in Mastomys natalensis in Guinea, West Africa. Vector Borne and Zoonotic Diseases. 2007; 7(2):119-28. Epub 2007/07/14. <https://doi.org/10.1089/vbz.2006.0520> PMID: 17627428.

Reviewers' comments:

Reviewer #1 (Remarks to the Author):

This manuscript focuses on the impact of non-native rodent species (*Rattus rattus*) on the ecology of native *Mastomys N.*, and how this affect spillover of Lassa fever. I read the manuscript with interest; Assessing ecological changes in the reservoir is extremely important and the results are certainly intriguing. I think the authors did a good job in terms of fieldwork and analysis, but I believe that they need to improve the text, especially in the methodology session.

Thank you for your positive assessment. Your comments have helped us improve the clarity of the manuscript, particularly in the Methods section, as we explain in detail below.

Below are my comments:

- **Methods:** I would really appreciate if the authors can explicitly write down the equations of the Poisson model. I know that many researchers just explain the model in a narrative way rather than mathematical equations, but I don't think that this is a good practice. It makes the methodology less transparent and hard to follow. Please write down your equations and explicitly relate your assumptions with the equations. You can do this in the supplementary material.

We have added a mathematical presentation of our model as Figure S1 in the Supplemental Information. In addition, all of our Stan model code is openly available on GitHub: https://github.com/eveskew/rat_invasion/tree/main/stan_models. Stan code is quite similar to mathematical equations and should provide further transparency for interested readers.

- I presume that both *Mastomys N.* and *Rattus Rattus* exhibit seasonal patterns. If so, are the pattern of seasonality the same for both species? Shouldn't these be included in the analysis? Please explain.

It is true that *Mastomys natalensis* abundance and behavior might vary seasonally and could therefore influence trap success. This could cause problems in our analysis of the *Rattus rattus* presence effect if, for instance, certain sites were only sampled during certain seasons. To control for this issue, and following prior literature on *Mastomys natalensis* ecology, we now incorporate an effect of seasonality (wet vs. dry season) in all models throughout the paper. In essence, this main effect allows *Mastomys natalensis* catch per trap to vary according to the sampling season. The inclusion of seasonal effects in our models required the reorganization of our trapping data into visit-level data (information on the sampling season of specific site visits is lost when aggregating trapping data at the site-level).

See lines 137-143 in the manuscript:

“While our primary aim was to understand the effect of *R. rattus* on *M. natalensis* captures, we also wanted to account for potential seasonal effects on *M. natalensis* abundance and

activity that could influence capture success [26, 34]. As such, we first organized the trapping data at the visit-level, with each observation representing trapping results from a given site visit (n = 71 unique site visits that involved house trapping) and each visit coded as having occurred in the wet season (May-October) or dry season (November-April) [26].”

Although seasonality effects were statistically important in some cases, they were always of lesser magnitude than the *Rattus rattus* presence effect, supporting our broad argument about the key influence of *Rattus rattus* on *Mastomys natalensis*.

Finally, because the *Rattus rattus* predictor variable was coded at either the site- or house-level in all models (i.e., a specific site or a specific house was coded as *Rattus rattus* present if the species had ever been detected there), seasonal effects on *Rattus rattus* were not relevant to our specific modeling framework.

- Line 88-90, when the author say Sampling that generated the Guinea dataset was generally similar, except trapping effort was targeted within homes, in cultivated areas, and in forests. Please explain why the sampling in Guinea was done slightly different.

Sampling was done differently in Guinea simply because our manuscript aggregates data from two distinct field sampling efforts conducted by two different field teams. However, in response to comments from Reviewer #2, we have reorganized the manuscript’s main text analyses to focus exclusively on house trapping data, which are directly comparable between Guinea and Sierra Leone.

- Line 111-113, See point above. Explicitly write down in mathematical terms the rate of the Poisson process, intercept and offset terms along with their interpretation.

The addition of Figure S1 to our Supplemental Information and the associated figure caption should address these concerns.

- Line 111-113, please explain the rationale to model the effect of *Rattus Rattus* in terms of presence/absence alone and not their abundance (I guess you can catch 1-2 *Rattus Rattus* at most, so your choice is reasonable to me, but I think you need to explain this).

In lines 143-147, we now explain that:

“Some site visits did not result in *R. rattus* capture even when *R. rattus* was known to occur at the site based on other visits. Therefore, in our models, we opted to formulate the *R. rattus* effect as a site-level binary variable that indicated whether *R. rattus* had been captured at the site during any site visit.”

In other words, with our visit-level data aggregation strategy, we did not want to code certain site visits as *Rattus rattus* negative if *Rattus rattus* was in fact detected at that same site during other site visits. Thus, we generated a binary, presence/absence *Rattus rattus* variable using all trapping data available from a given site.

- Line 122-124. If the mean was 0.077, I am not sure if a median of 0.044 is compatible with a Poisson distribution (there are some approximations in the literature). Have the authors checked that the empirical data qualitatively follow a Poisson distribution?

The core issue here concerns the prior for a single parameter in our model (the intercept/grand mean parameter). We have added a supplemental figure to our manuscript, Figure S2, to help illustrate how our informative prior, Normal(-3.1, 1.1), implies a Poisson rate distribution with the summary statistics we report in the manuscript (mean = 0.082, median = 0.045).

We believe the secondary question here stems from a common misconception: the linear modeling framework does not assume that the raw outcome variables match in a direct way with the outcome distribution of the model. Rather, the assumption is that after conditioning on all predictor variables, the residuals from the model are reasonably described by the outcome distribution. McElreath 2020, *Statistical Rethinking*, page 314 provides a nice explanation using the example of a Gaussian model (i.e., standard linear regression):

“...at most what a Gaussian likelihood assumes is not that the aggregated data look Gaussian, but rather that the *residuals*, after fitting the model, look Gaussian. So for example the combined histogram of male and female body weights is certainly not Gaussian. But it is (approximately) a mixture of Gaussian distributions. So after conditioning on sex, the residuals may be quite normal.”

In our case, we have chosen the Poisson outcome distribution since it fits naturally with the data type of our empirical data (non-negative integer counts of *Mastomys natalensis*), not because of the distribution of the empirical data.

- Line 128 and line 131. I can see why mean of zero, but why standard deviation of 1?

Put simply, this prior was chosen as a conservative prior for the effect of *Rattus rattus* presence on *Mastomys natalensis* catch per trap.

The most important feature of this prior is just that it is centered at 0. Thus, the *a priori* knowledge of the model is that the effect of *Rattus rattus* presence is equally likely to be negative or positive. Given that the prior should have a mean of 0, the only other decision is choosing the magnitude of the standard deviation. Unfortunately, the default “flat” priors often used in Bayesian analysis software may have conditioned ecologists to accept the use of vague priors, such as Normal(0, 10), that can have serious unintended consequences when used within generalized linear modeling frameworks (McElreath 2020, *Statistical Rethinking*, page 348-352; Banner et al. 2020, “The use of Bayesian priors in ecology: The good, the bad and the not great”). The issue is that, due to the link functions used within Poisson and binomial regression models, vague priors can imply absurdly small or large values on the outcome scale of interest.

To illustrate the consequences of our prior choice, recognize the fact that for all normal distributions, ~95% of the probability mass lies within 2 standard deviations on either side of the mean. Therefore, for the sake of argument, we can use the rough rule of thumb that a normal distribution views any value within 2 standard deviations of the mean as reasonably likely. Now, using only a point estimate rather than a distribution for ease of explanation, assume that the fit intercept parameter in our Bayesian model is -2. This implies a *Mastomys natalensis* catch per trap of:

$$\exp(-2) = 0.135 \text{ Mastomys natalensis per trap}$$

when *Rattus rattus* is absent.

What does our choice of the Normal(0, 1) imply about how the model views the likely magnitude of the *Rattus rattus* effect, *a priori*? Here, we can use the 2 standard deviation rule of thumb. Before seeing any data, this prior implies that the model roughly considers anywhere from -2 to 2 as reasonable values for the *Rattus rattus* effect. How would these values affect our understanding of *Mastomys natalensis* catch per trap in the presence of *Rattus rattus*? To do understand this effect, we need to add the *Mastomys natalensis* effect into our prior calculation.

Taking a low prior estimate for the *Rattus rattus* effect of -2:

$$\exp(-2 - 2) = \exp(-4) = 0.018 \text{ Mastomys natalensis per trap}$$

Taking a high prior estimate for the *Rattus rattus* effect of 2:

$$\exp(-2 + 2) = \exp(0) = 1 \text{ Mastomys natalensis per trap}$$

Thus, we see that our prior for the *Rattus rattus* effect implies that *Rattus rattus* presence could change *Mastomys natalensis* catch per trap by almost an order of magnitude in either direction (from ~0.1 to ~0.02 or from ~0.1 to ~1). Ultimately, our argument is that this choice of prior constrains the *Rattus rattus* effect to be a substantial, but biologically reasonable, effect.

In contrast, imagine this exact same illustrative example but with a vague prior of Normal(0, 10) for the *Rattus rattus* effect. This prior would imply, *a priori*, that we believe the *Rattus rattus* effect might reasonably be anywhere between -20 and 20 (2 standard deviations on either side of the mean). Translated to our outcome scale of interest, this prior would imply estimates of *Mastomys natalensis* catch per trap in the presence of *Rattus rattus* ranging from:

$$\exp(-2 - 20) = \exp(-22) = 0.000 \text{ Mastomys natalensis per trap}$$

to

$$\exp(-2 + 20) = \exp(18) = 65,659,969 \text{ Mastomys natalensis per trap}$$

These numbers are no mistake! Using a prior of Normal(0, 10) for the *Rattus rattus* effect would imply that we believe, *a priori*, that the presence of *Rattus rattus* alone might change *Mastomys natalensis* catch per trap from 0.135 animals per trap to 65,659,969 animals per trap. Clearly this is an absurd assumption, but these are the dangers of adopting flatter priors when using regression models with link functions.

In sum, we argue that our choice of the Normal(0, 1) prior constrains the biological effects in our models to be of a plausible magnitude. These are more conservative choices than are typically made by those using analysis software defaults.

• Line 136-144 and 188-195. I believe it is standard practice to show some diagnostic of the MCMC, you can do this in the supplementary materials.

We now include MCMC trace plots for fit models in the Supplemental Information.

• Line 162-171. I must admit that I am not clear what the authors exactly did. For example:
- when they say rows equal to the total number of houses, do they refer to total number of houses for different sites or what?

We have clarified on lines 219-222:

“The outcome data for this analysis was a *M. natalensis* detection matrix (1/0; detected/not detected) with rows equal to the total number of houses (n = 560) and columns equal to the maximum number of consecutive trapping nights at a house (n = 4).”

- I am not clear if they did a spatial analysis, if so, do they take into account spatial clustering?

The occupancy analysis is not spatially explicit. Rather, each household is treated as a distinct sampling unit nested within a village unit.

- Do they treat each occupancy as Bernoulli process?

Yes, occupancy models treat the latent presence/absence variable as Bernoulli distributed. The primary citation we use to support our occupancy analyses (Doser et al. 2022, “spOccupancy: An R package for single-species, multi-species, and integrated spatial occupancy models”) usefully summarizes the large literature on occupancy modeling, including mathematical equations.

- Is this single species or multispecies?

We have clarified in text on lines 215-217:

“As such, these house-level data were amenable to a single-species occupancy analysis focused on the occupancy status of *M. natalensis*.”

- What are the predictors? (I guess the nightly trapping effort and cumulative *M. natalensis* caught in the house but not clear).

The predictors identified by the reviewer are for the detection component of the model only. Lines 222-229 explain that:

“Occupancy models allow distinct predictor sets to be used for the occupancy and detection components of the model. For the occupancy component of the model (i.e., those factors predicting whether or not *M. natalensis* actually occupies a given house), we used an intercept term, a main effect of *R. rattus* presence/absence at the site (village), and a main effect of sampling season (wet vs. dry season). In addition, we included a site-level (village-level) varying intercepts structure. In essence, the predictor structure for the occupancy component of the model mirrors the predictor structure for the house-level hierarchical Bayesian model described above.”

We also highlight that we’ve worked to make our openly available code easy for others to navigate. For example, our house-level modeling script (https://github.com/eveskew/rat_invasion/blob/main/scripts/06_house_level_analyses.R) includes the model-fitting code for the occupancy model in question:

```
# Fit an occupancy model
out <- spOccupancy::PGOcc(
  occ.formula = ~ Rra_at_site + wet_season + (1|site_numeric),
  det.formula = ~ trap_count + cumulative_Mna_count,
  data = data.list,
  priors = priors.list,
  n.burn = 10000,
  n.samples = 35000,
  n.chains = 4,
  verbose = TRUE
)
```

We hope this makes all modeling choices as accessible and as transparent as possible.

• Line 280-281. When they say the estimated effect of *R. rattus* presence at a site was -1.12 [-2.90, 0.74]. I am not clear what the numbers exactly measure (although the interpretation is ok). Same when they say and posterior overlaps with 0 in the 99% HPDI, posterior of what? Again, writing equations would massively help.

We hope the inclusion of Figure S1 and the availability of model code on GitHub helps to clear up any confusion. The estimate mentioned is the estimated coefficient for the *Rattus rattus* presence/absence predictor variable. The posterior being referred to is the posterior distribution for that same coefficient parameter. The numbers given are a summary description (mean and HPDI) of the full posterior distribution for the parameter. To understand what this effect actually means on the outcome scale of interest (i.e., how does this estimate translate into an effect on *Mastomys natalensis* catch per trap?) requires

transformation through the model's inverse link function, as we illustrated in the example calculations above in response to another reviewer comment. This is also why we made the effort to translate fit model parameters into meaningful predictions on the outcome scales of interest (e.g., showing distributions for predicted *Mastomys natalensis* catch per trap) in all of the manuscript's figures.

Reviewer #2 (Remarks to the Author):

This paper provides a contemporary analysis of a mix of self and published data on rodent trapping in Sierra Leone and Guinea in the context of LFV spill over risk - focused on the role of invading *R. rattus*. The authors I believe correctly identify the need for community rodent ecology research in this area and the importance of understanding shifting habitat and population dynamics across LFV infected landscapes, villages and towns. They attempt to use available data and some self collected material to prove the point but I fear this is a step too far. The use of fancy statistical methods and modelling will not solve this. More systematic trapping methods are the only way to get to the truth and across the whole ecological and epidemiological landscape for LFV. My recommendation is to reduce the scope of what the paper is attempting to do and focus where the comparisons between own and other data are valid. I see this probably only in the trapping in housing and view on *M. natalensis* and *R. rattus* in this context. There is no crime in speculation in discussion on the core subject but to make the sort of statements suggesting that invasion of *R. rattus* might be good outcome is surprising given the isolated issue examined. I currently cannot accept this publication with extensive revision although I see a lot of merit in it so would encourage a more careful scope of work. I have included some specific comments on the PDF for your consideration.

We agree that more systematic rodent trapping is necessary to fully understand the risk landscape for Lassa fever across West Africa. However, we believe our current efforts are a key step in that direction. In particular, we highlight the fact that researchers have speculated about the negative effects of *Rattus rattus* on *Mastomys natalensis* for decades. We set out to test this existing hypothesis with a dataset that we felt could be particularly informative. We further note that all analyses in our paper support the idea that *Rattus rattus* has a negative influence on *Mastomys natalensis*. Thus, we find support for the *Rattus rattus* competition hypothesis, regardless of the specific dataset, data aggregation strategy, predictor variable construction, or modeling framework we use.

We are thankful to the reviewer for highlighting issues with the paper's scope. In response to these concerns, we have reorganized the main text of the manuscript to focus exclusively on house trapping data. These are the traps that are most directly comparable between the two field sampling efforts in Sierra Leone and Guinea (since outside trapping efforts in those countries varied in their trap placement) and these are the traps that are presumably the most relevant for Lassa virus ecology and transmission (since much human exposure to infected *Mastomys natalensis* presumably takes place indoors). As a result, we believe the main text of the manuscript is now more streamlined since the visit-level, house-level, occupancy, and LASV-infected rodent analyses all rely on data from traps placed within houses. We still include the analysis of the full dataset with all traps in the Supplemental Information (again, we emphasize that this analysis also shows a negative effect of *Rattus rattus* on *Mastomys natalensis*).

Finally, we have tried to take care to maintain nuance in our discussion of the *Rattus rattus* invasion. While we do present evidence that *Rattus rattus* invasion is reducing Lassa virus spillover risk, this certainly does not mean that *Rattus rattus* invasion is a universal good. In lines 486-488 we now state:

“We stress that our findings are specific to the Lassa fever system and that invasive *Rattus* do host other zoonotic pathogens that can threaten human health [14, 70].”

Line 30: This statement on burden is premised on underreporting which is very likely to occur simply due to the weak health systems in the region. But clinical disease and death rate underreporting may or may not be true. The Simon's paper does not actually confirm this either way and states uncertainty but showing that underreporting of cases is likely - we simply don't know. Most cases might simply recover without hospitalisation. It has been a standard dogma for years but given recent findings of very high seroprevalence in Sierra Leone it may well be true that infection rates are high and that case rates and infection fatality rates may be very low. Case fatality rates are high. It would be good now that more concrete data is arising on incidence and prevalence that a more nuanced introduction is developed, reflecting this advance in knowledge. Estimates on disease burden are very crude and dated but the statements do not reflect this. Please update and provide a balanced view.

Our goal was to write a punchy Introduction section that quickly proceeds towards explaining the key connections between rodent ecology and Lassa virus spillover risk. The point we seek to make in this sentence, measurement uncertainty aside, is simply that Lassa virus is a major public health issue in West Africa. Thus, we have dropped the mention of underreporting entirely:

“Estimates suggest that Lassa virus infects hundreds of thousands of people and causes thousands of deaths annually [1, 5, 6].”

While the numbers we cite are broad ranges, we note that they are consistent between relatively old studies (McCormick et al. 1987) and more recent, sophisticated modeling studies that reached similar conclusions regarding Lassa burden across West Africa (Basinski et al. 2021).

Line 31: and vectoring? a statement on this would be useful with latest publication on the subject.

We have now bolstered this statement with three citations that all mention potential alternative hosts of Lassa virus. However, none of these papers specifically use the terminology of “vectors” or “vectoring”. Indeed, we are unaware of strong evidence (like the viral phylogenomic studies mentioned in our next comment) that directly link Lassa infection in these alternative hosts to Lassa infection in humans. Thus, we intentionally limit our statement to suggest that alternative hosts may help to maintain Lassa within the rodent community but stop short of addressing any potential role they may play in animal-to-human Lassa virus transmission.

Line 32: Evidence suggests....but data is fairly sparse on this aspect so perhaps a little less certainty would reflect the state of knowledge.

We find that the existing evidence in the literature makes a convincing case that most Lassa transmission is zoonotic. Multiple lines of evidence support this assertion. First, a modeling

study which we cite that explicitly set out to disentangle animal-to-human vs. human-to-human transmission in Lassa estimated that ~20% of cases were human-to-human. That implies most Lassa transmission (80%) remains zoonotic. Further, the two viral phylogenomic studies we cite provide complementary evidence that human Lassa isolates cluster with rodent-derived viral isolates, as would be expected given frequent rodent-to-human viral transmission. Andersen et al. 2015 conclude: “most human LASV infections represent independent transmissions from a genetically diverse reservoir.” Kafetzopoulou et al. 2019 conclude: “Even when applying liberal assumptions for the number of mutations during human-to-human transmission, the vast majority of cases during the 2018 outbreak resulted from spillover from the natural reservoir.” Thus, we feel that our statement conforms with the strongest available evidence.

Line 43: this invasion is also driven by settlements providing suitable peridomestic and agricultural environments with storage and food sources for this species. Some ecology on this would be welcome. It is not happening just because of rat movements as suggested by this statement.

In lines 52-55 we now state:

“As a result, the black rat was likely established in some inland areas of West Africa by the middle of the 20th century [18, 20, 22]. However, invasion across the region is still ongoing and may be facilitated by increasing availability of anthropogenic food subsidies and man-made structures that serve as rodent habitat [21, 23].”

Line 44: potentially

Text changed to “potentially significant”, as suggested.

Line 56: Some data on the potential role of *R.rattus* as a maintenance host or vector for LFV would be useful here as no mention is made of this possibility. Please provide available data and references. Also a more comprehensive critique of this would be sensible given the significant role of *R. rattus* in a range of other zoonoses. Some publications (see Simons et al 2023) look at zoonotic risks across available sample data for WA rodents please include. It seems unlikely this would be a positive health intervention. A mention of the introduction of domestic cats in this regard would also be useful and how this has affected LFV if at all.

Our goal in this Introduction paragraph was to briefly summarize the history of the “*Rattus rattus* hypothesis” in the Lassa virus literature. Thus, in this section we do mention the idea of *Rattus rattus* introduction as a form of biocontrol, not by way of endorsement (we definitely do not endorse this idea) but simply as historical scientific context.

We believe the Reviewer’s other concerns are addressed in other portions of the manuscript.

For example, we mention the potential role of *Rattus rattus* as a Lassa virus host in the Methods:

“While studies have occasionally presented evidence of Lassa virus infection in rodents identified as *R. rattus* [46], these have been interpreted as rare, transient cross-species transmission events [26], and researchers generally report a lack of infection in this species, particularly in our study region of Sierra Leone and Guinea [7, 8, 47].”

We stress the role of *Rattus rattus* as a zoonotic host in the Discussion where we have now incorporated the Simons et al. 2023 citation:

“While local people in invaded regions may still seek to eliminate *R. rattus* populations because of their activity as household pests [23], our results show that black rats actually seem to reduce Lassa exposure, a counterintuitive ecosystem service provided by a non-native species [67–69]. We stress that our findings are specific to the Lassa fever system and that invasive *Rattus* do host other zoonotic pathogens that can threaten human health [14, 70].”

Line 66: This statement is in the wrong place by all means mention the objective of your research and the paper but leave results to results.

We have modified this portion of the Introduction to simply mention the objectives of our research without any reference to our specific study findings.

Line 68: I am concerned about possible flaws in the approach and analysis. There are inherent assumptions in this approach that all spill over occurs in village settings and immediate surroundings whilst there is evidence that spill over could occur in the agricultural settings, on fields and some distance from the villages. This agricultural area is not included in the analysis and results may consequentially be biased and spurious to some extent without a comprehensive epidemiological view.

The modelling is also dependent on a lot of statistical manipulation to iron out a fundamentally variable data set as each trapping was not standard or used for the purposes of this research. I am concerned that this again is putting some potential bias into the process and is made to fit rather than fitting the truth.

In response to the Reviewer’s other comments, we have revised our manuscript’s main text to focus solely on house trapping analyses. We hope that this more restrictive study scope provides a stronger, more streamlined presentation.

We emphasize that we have worked to make our statistical approach as transparent as possible on our project’s GitHub repository page. We provide all model code and data processing scripts so that interested readers can interrogate our analyses. However, we also emphasize that we consistently find a negative effect of *Rattus rattus* on *Mastomys natalensis* in multiple different analyses. We therefore view the *Rattus rattus* effect we recover as quite robust to modeling choices and frameworks.

Line 145: I am more comfortable with the house level analysis and it might be sensible to focus on this only rather than the earlier transect data. If the study is restricted to this showing displacement of *Mastomys* from the house setting it will enable a more specific conclusion on spill over risk in housing in infected zones. This is my recommendation.

Thank you for this recommendation. As indicated in our response to this Reviewer's general comments, we have revised the manuscript's main text to focus solely on house trapping analyses.

Line 199: If the historical and contemporary sampling and reporting was standardised (especially habitat classification) and comparable I would agree but this is not the case and I think a stretch too far in analysis.

We have removed the historical vs. contemporary sampling comparison in our revised manuscript in favor of focusing exclusively on contemporary data.

Line 202: but not the same.

We have removed the historical vs. contemporary sampling comparison in our revised manuscript in favor of focusing exclusively on contemporary data.

Line 212: this will help and might get you off the hook - and reflects my earlier advice on how you use the data you have and the scope of analysis.

Thank you. While we did attempt to make this historical vs. contemporary sampling comparison as strong as possible by selecting the datasets carefully, we ultimately decided to remove this analysis in favor of focusing exclusively on contemporary data.

Line 226: this may or may not be a key factor in *M. natalensis* infection prevalence - it is likely but other factors may well be at play and this is too simplistic a view - density might well lead to high infection rates initially but lower maintenance levels of infection as immunity in the population rises. It is when there is naivety in the population that spill over risk might be higher as a wave of virus spreads through rodent populations. So unstable *Mastomys* populations, along with other rodent community might well be critical to viral load, due to seasonal factors e.g. crop production when some rodent migration occurs and occupancy changes across the village and rural systems.

It is true that our spillover index does not account for within-host dynamics such as varying viral loads within infected rodents. Nevertheless, we argue that we have generated the most straightforward, defensible risk index that is possible given the data at hand: a count of Lassa-positive (sequence-confirmed) *Mastomys natalensis*.

Line 231: given the lack of data on this subject I do not think it is in fact a reasonable assumption.

We agree that more work needs to be done on alternative Lassa virus host species, but given: 1) the general acceptance of *Mastomys natalensis* as the primary host of Lassa virus, 2) the consistent association of Lassa virus with *Mastomys natalensis* through decades of field studies, 3) the fact that *Mastomys natalensis* are highly commensal, and 4) viral phylogenomic papers that link rodent and human Lassa cases (Andersen et al. 2015, Kafetzopoulou et al. 2019), we think it is reasonable to treat *Mastomys natalensis* as the primary driver of human Lassa virus exposure within the household environment.

Line 234: this is all very wooly and suiting the presumptions rather than showing good science. Rather state the facts and let the reader conclude.

We have modified the sentence to read:

“While studies have occasionally presented evidence of Lassa virus infection in rodents identified as *R. rattus* [46], these have been interpreted as rare, transient cross-species transmission events [26], and researchers generally report a lack of infection in this species, particularly in our study region of Sierra Leone and Guinea [7, 8, 47].”

We believe this is an accurate and fair accounting of the literature:

- Wulff et al. 1975 did report Lassa infection in *Rattus rattus* but openly discuss the possibility that they misidentified *Mastomys natalensis* as *Rattus rattus*: “With the difficulties inherent in rodent classification under field conditions, the possibility that misidentifications may have occurred must be strongly considered.”
- Demby et al. 2011, who do report serology-positive *Rattus rattus*, also give strong caveats in their Discussion section and explicitly cast any infections in this species as potentially transient: “Despite occasionally finding Ab or Ag in other species, we were able to grow LV only from *Mastomys*. This result likely reflects the possibility of non-*Mastomys* to be occasionally and transiently infected with LV, so-called spillover infections. The isolation of LV from non-*Mastomys* has been reported in a single study. However, the authors readily brought up the possibility of species misclassification (Wulff et al. 1975). It is also possible that positive Ab and Ag results from non-*Mastomys* represent cross-reactions with other, non-Lassa arenaviruses (Wulff et al. 1977, Johnson et al. 1981, Gonzalez et al. 1983, Swanepoel et al. 1985).”
- Finally, we end by citing a series of three studies from Sierra Leone and Guinea that fail to recover Lassa virus sequence from *Rattus rattus*.

Line 242: which is again why I think the paper should only focus in this area of research. It will not be a lost cause then.

Thank you for this recommendation. As indicated in our response to this Reviewer’s general comments, we have revised the manuscript’s main text to focus solely on house trapping analyses, including our Lassa virus spillover risk analysis which only uses rodent trapping and viral data collected from within homes.

Line 244: more assumptions ...increasing data showing wider host range should be taken into account...

We have followed the Reviewer’s recommendation to focus exclusively on house trapping data, which they judged to be the strongest portion of our data for analysis.

Line 302: no effort is made in trying to determine if the historical habitat has changed substantially whether in village size structure housing etc which may be playing a role in this apparent although weakly correlated finding.

We agree it is quite difficult to get a strong comparison of historical and contemporary rodent sampling given the various factors that may have changed over the decades. Thus, we have removed the historical vs. contemporary sampling comparison in our revised manuscript in favor of focusing exclusively on contemporary data.

Line 317: these results maybe, with all their flaws and I am not comfortable with the proposed certainties.

We have attempted to make our methods and statistical procedures as transparent as possible. We note that motivated readers can find our data, model code, and scripts openly available on the project’s GitHub repository for inspection.

Line 327: not convinced I am afraid

We hope that our revised manuscript, which focuses exclusively on house trapping data and shows a negative effect of *Rattus rattus* on *Mastomys natalensis* across all analyses, including:

- **A visit-level analysis of house trapping data with a site-level *Rattus rattus* predictor**
- **A house-level analysis of house trapping data with a site-level *Rattus rattus* predictor**
- **A house-level analysis of house trapping data with a house-level *Rattus rattus* predictor**
- **An occupancy analysis of house trapping data with a site-level *Rattus rattus* predictor**

will be more convincing to the Reviewer.

Line 343: this may simply reflect urbanisation as indeed seems to be associated with lowered LFV risk whether rat related or not remains to be seen.

Correct. We offer alternative explanations for this pattern in our Discussion:

“The absence of *M. natalensis* from western, coastal Guinea has been noted previously, and this gap in the distribution may be a natural range limit or a consequence of replacement by the congeneric species *M. erythroleucus* [8,27,28]. However, our data suggest this distribution pattern may also be consistent with exclusion by *R. rattus* [32], which is abundant at many of these same coastal sites.”

Line 391: I feel this is a premature statement and potentially highly misleading and might lead to increased risk if advocated for zoonosis more generally.

We have added an explicit caveat to address this concern:

“While local people in invaded regions may still seek to eliminate *R. rattus* populations because of their activity as household pests [23], our results show that black rats actually seem to reduce Lassa exposure, a counterintuitive ecosystem service provided by a non-native species [67–69]. We stress that our findings are specific to the Lassa fever system and that invasive *Rattus* do host other zoonotic pathogens that can threaten human health [14, 70].”

Line 414: I welcome your self criticism in this last paragraph and I would hope this would cool down the degree of certainty that creeps in to the narrative in this piece of work.

Thank you for your comments throughout. We hope that our response to your suggestions has made for a more robust, balanced manuscript.

Reviewer #3 (Remarks to the Author):

Summary

In this work, the authors use rodent trapping data from 28 sites in Sierra Leone and Guinea to evaluate the impact of the presence of the invasive species *Rattus rattus* on the density of *Mastomys natalensis*, a rodent species considered the primary host of Lassa virus. By building hierarchical models based on house-level and site-level data, as well as an occupancy model based on house-level data. They identify a negative effect of *R. rattus* presence on (1) *M. natalensis* catch per trap numbers both at site- and house-levels (hierarchical models), (2) *M. natalensis* probability of detection (occupancy model), (3) number of catch per trap of Lassa-positive *M. natalensis* -used here as an indicator of Lassa virus spill-over risk.

General Comment

This interesting work adds to previous studies that have started to describe the impact of *R. rattus* presence on *M. natalensis* in Guinea, Benin, and Niger, and provides a first assessment of a potential effect on Lassa spill-over risk. There are, however, several limitations, both in the analyses and conclusions, that are of concern at this stage.

Thank you for your kind assessment of our manuscript and suggestions for further improvement.

Major Comments

Results Section 1: Contemporary sites with *Rattus rattus* have fewer *Mastomys natalensis*

-Confounding variables:

In their hierarchical model, the authors account for the effect of several variables on catch per trap numbers including *R. rattus* presence, total trap-nights number, heterogeneity of sampling efforts across sites. However, Several other factors (environmental, temporal) are known to affect *M. natalensis* populations and in particular, seasonal variations in population numbers and localisation (e.g. in [1]). I would recommend accounting at least for this possible confounding variable in the analysis.

Thank you for raising this issue and highlighting this relevant citation. In response to this concern (and a similar one raised by Reviewer #1), we have revised all analyses in the manuscript to include a seasonality effect (wet vs. dry season sampling) on *Mastomys natalensis*. While seasonality does have an effect on *Mastomys natalensis* catch per trap in some analyses, it is always of lesser magnitude than the *Rattus rattus* presence effect.

-Specificity of the effect:

The main point of this study is to demonstrate a specific (and negative) effect of *R. rattus* on *M. natalensis* presence. While the authors do identify an effect of *R. rattus* on *M. natalensis*, it is not clear that this effect is specific to *R. rattus*. Performing similar comparisons using a “control”

population (another rodent species present at study sites) would demonstrate the specificity of the *R.rattus*/*M.natalensis* antagonism.

Thank you for this insightful comment. To address this issue, we have now fit alternative models that consider the presence of other rodent community members in place of *Rattus rattus*. We chose to evaluate the effect of any rodent that had more than 100 captures in the full trapping dataset. While this is an arbitrary cutoff, it means we evaluate key rodent community members that are roughly similar in abundance to *Rattus rattus* (which had 151 captures in the full dataset). Overall, we evaluated four additional rodent species: *Mastomys erythroleucus* (377 captures), *Praomys rostratus* (167 captures), *Mus mattheyi* (135 captures), and *Praomys daltoni* (114 captures).

The results of these model-fitting exercises are presented in the Supplemental Information, Figure S4. While the mean estimated effects of *Mastomys erythroleucus* and *Praomys rostratus* were both negative, the posteriors for these effects overlapped with 0 much more than the analogous effect for *Rattus rattus*, suggesting greater uncertainty about the direction of the effect in these two alternate species. In contrast, the estimated effects for *Mus mattheyi* and *Praomys daltoni* both had positive means (and again rather uncertain posterior distributions overall). In sum, we take these additional results to support our case that *Rattus rattus* presence has a uniquely strong negative effect on *Mastomys natalensis*. These new findings are explicitly discussed in the main text, lines 332-336:

“The negative effect of *R. rattus* on *M. natalensis* was unique within the sampled rodent community. Estimates of the effect of four alternative rodent species on *M. natalensis* all overlapped with 0 in the 99% HPDI and even overlapped 0 in the much more conservative 80% HPDI (Fig. S4). As such, models did not suggest a strong directional influence of these species on *M. natalensis* captures, as was the case for *R. rattus*.”

-Impact of the prior:

The authors describe using an informed prior that reflects pre-existing evidence regarding *M. natalensis* catch per trap. Did the authors evaluate how sensitive their estimates were to this choice of prior?

In response to this comment, we investigated the use of other priors for the intercept parameter in our visit-level Bayesian model fit to house trapping data. We assume alternative priors of interest would be uninformative priors naively centered at 0 (although we note this approach can be dangerous with Poisson models since all prior mass < 0 becomes bunched between 0 and 1 on the outcome scale and large negative prior values translate to extremely small values on the outcome scale; McElreath 2020, *Statistical Rethinking*, page 349). Key parameter estimates from models using alternative priors are as follows:

Intercept prior of Normal(0, 1) -> Intercept estimate: -2.90 [-4.18, -1.72]; *Rattus rattus* estimate: -2.47 [-4.01, -0.95]

Intercept prior of Normal(0, 2) -> Intercept estimate: -3.54 [-5.32, -2.18]; *Rattus rattus* estimate: -2.01 [-3.60, -0.22]

Intercept prior of Normal(0, 3) -> Intercept estimate: -3.75 [-5.86, -2.27]; *Rattus rattus* estimate: -1.88 [-3.55, -0.04]

Compared with our informative intercept prior of Normal(-3.1, 1.1) -> Intercept estimate: -3.68 [-5.32, -2.38]; *Rattus rattus* estimate: -1.93 [-3.52, -0.24]

As we can see, there is a relationship between the intercept prior assumed and the posterior for the *Rattus rattus* effect: as the intercept prior becomes flatter (i.e., larger standard deviation) it allows for increasingly negative posterior values of the intercept parameter. More negative intercept estimates, which imply smaller baseline *Mastomys natalensis* catch per trap values, in turn begin to erode the signal of *Rattus rattus* presence on *Mastomys natalensis*. What we're seeing here is essentially a floor effect: when the model thinks baseline *Mastomys natalensis* catch per trap in the absence of *Rattus rattus* is extremely close to 0, *Rattus rattus* presence necessarily has less of an impact on *Mastomys natalensis* catch per trap since catch per trap can't go below 0 in a Poisson model.

Most importantly, however, we highlight the fact that all of these alternative priors still return a negative effect of *Rattus rattus* on *Mastomys natalensis* catch per trap. This result, with varying degrees of precision, was also found across all models we tested in our paper. Ultimately, we argue that our decision to use an informed prior was the most appropriate. Indeed, this is the only choice that allows us to leverage the unique benefits of the Bayesian modeling approach by incorporating knowledge gained from previous scientific investigations into our own.

Results Section 2: Comparison of contemporary and historical sampling for *Mastomys natalensis*

In this section, the authors state that they “recovered some evidence for temporal declines in *M. natalensis* density in Sierra Leone’s eastern province” however, model estimates of average site-level *M. natalensis* catch per trap are close with broadly overlapping ranges (0.033 [0.019, 0.054] historically - 0.024 [0.007, 0.050] currently) and an 99% HPDI for the effect of contemporary sampling that overlaps with 0. Given the weakness of the evidence, I would recommend a more cautious formulation of the results.

We agree that the comparison of historical and contemporary *Mastomys natalensis* sampling in Sierra Leone was not the strongest element of our original manuscript. Partially, this was because there were only a limited number of study sites where meaningful temporal comparison could be made. In addition, as we speculated in our previous Discussion section, this may have been because *Rattus rattus* was already present in Sierra Leone even during the historical sampling period. Thus, the temporal comparison may not have adequately captured the state of the rodent community in Sierra Leone before and after *Rattus rattus* invasion. All these facts considered, we have removed this analysis from our paper and have decided to focus solely on contemporary trapping data.

Results Section 3: Contemporary sites with *Rattus rattus* have reduced Lassa virus spillover risk

As for section 1, this part of the analysis would be much stronger if account for seasonal variations and assessing the specificity of the effect of *R. rattus*.

This analysis now accounts for seasonal effects on *Mastomys natalensis* activity, as described in our response to Reviewer #1's comments. As mentioned above, we test for the specificity of the *Rattus rattus* effect in the larger visit-level dataset of *Mastomys natalensis*.

Minor comments

l43: "while invasion is ongoing in other locations [20]" Can you specify which other locations? In the current form, it is unclear which areas are the ones where invasion is ongoing (and which areas are the ones where *R. rattus* is established)

Good point. Our intention here was simply to signal that *Rattus rattus* invasion is ongoing across West Africa. Even within a single country, certain villages may already be invaded whereas others do not apparently have *Rattus rattus* (as our data show). In many parts of West Africa, the stage of invasion may be unclear due to data limitations. We have revised (lines 52-55) to keep the general idea intact without specifying where exactly invasion is happening (since we don't know with much accuracy):

"As a result, the black rat was likely established in some inland areas of West Africa by the middle of the 20th century [18, 20, 22]. However, invasion across the region is still ongoing and may be facilitated by increasing availability of anthropogenic food subsidies and man-made structures that serve as rodent habitat [21, 23]."

l44: "a outstanding" perhaps the authors rather meant "an outstanding"

Thank you. Change made as suggested.

l116-117: "in effect recovering inference about an typical site in the model's intercept term" this sentence seems to need editing

Sentence modified to read "a typical site".

l392-393: "our results show that black rats may actually protect against Lassa exposure" I would recommend using "reduce" rather than protect against

Change made as suggested.

References

[1] Fichet-Calvet E, Lecompte E, Koivogui L, Soropogui B, Dore A, Kourouma F, et al. Fluctuation of abundance and Lassa virus prevalence in *Mastomys natalensis* in Guinea, West

Africa. *Vector Borne and Zoonotic Diseases*. 2007; 7(2):119–28. Epub 2007/07/14.
<https://doi.org/10.1089/vbz.2006.0520> PMID: 17627428.

REVIEWER COMMENTS

Reviewer #1 (Remarks to the Author):

The authors have addressed my comments. The additional analysis on seasonal effects and also the impact of other 5 rodent species make the findings more robust. Pleased to see that the codes are publicly available on GitHub. I only have a couple of minor points listed below:

Lines 367-370. I am not fully clear what the authors want to say. Please rewrite.

Figure S3 and S5. The season effect parameter change direction when they use house trapping data and all trapping data as discussed in line 341-343. I am wondering if this reflect the fact that the detection of *Mastomys N.* inside houses changes with season and in opposite directions compared to proximity field (see E. Fichet-Calvet et al., "Fluctuation of abundance and Lassa virus prevalence in *Mastomys natalensis* in Guinea, West Africa.," *Vector Borne Zoonotic Dis.*, vol. 7, no. 2, pp. 119–28, Jan. 2007, doi: 10.1089/vbz.2006.0520.). I am aware that in S5 the author used all trapping data (and not outside house data), but maybe the author could discuss this.

Figure S7 and S8 seems to have the same caption (both "only house trapping"), I presume one of them used all trapping data.

Reviewer #2 (Remarks to the Author):

Dear Authors

Thank you for taking comments positively and changing the manuscript constructively. I am happy that all my concerns have been adequately addressed. I will revise my view now to publish.

Reviewer #2 (Remarks on code availability):

I have checked a sample and they look fine.

Reviewer #3 (Remarks to the Author):

General Comment

The authors made substantial efforts to address the concerns raised by the reviewers regarding the first version of the manuscript. However, I still have several concerns regarding the content of the manuscript which are detailed below.

-Caution regarding the scope of the conclusions

I would recommend that the authors be extremely cautious with the formulation of the results regarding the reduction of LASV spill-over risk, especially as they could result in supporting worrying approaches of biocontrol that create more risks than they eliminate. Also - and although the authors do comment on the elements available to support the absence of role of *R. rattus* as a potential vector for LASV in the manuscript- it is challenging to assess the real risk of *R. rattus*-mediated LASV transmission as (1) very little field and experimental data is available and (2) the ability of LASV to adapt to this host - currently considered a dead end - is unknown. In light of these considerations I would recommend formulating this manuscript much more cautiously with regard to the scope of their results and adding a specific paragraph to detail the limitations of the study's conclusions in the discussion.

-Effect of seasonality

In the new version of the manuscript, the authors account for seasonality in their models by adding a specific binary predictor relative to wet season sampling, which only partly addressed my concerns on potential missing variables. The criteria used to discriminate between dry and wet seasons are not defined explicitly in the text which makes it difficult to assess the relevance of the new set of variables, this needs to be defined somewhere in the methods. Also, as with climate change, abnormal seasonal temperatures and precipitation are increasingly observed, I would recommend directly including environmental variables (temperature, precipitation, etc) that underlie the seasonal effect previously observed on *M. natalensis* populations to make the model more explicit and remove potential biases due to e.g., a wet season that would be abnormally dry and/or cold.

-Specificity of the effect

The authors controlled for the specificity of the effect of *Rattus rattus* on *Mastomy natalensis* populations at the site level by including other common rodent species as alternative predictors of *M. natalensis* captures. This correctly addressed my concerns regarding the site-level analysis; however, it would also be necessary to confirm the specificity of the effect of *R. rattus* on *M. natalensis* for the house-level and spill-over risk analyses.

REVIEWER COMMENTS

Reviewer #1 (Remarks to the Author):

The authors have addressed my comments. The additional analysis on seasonal effects and also the impact of other 5 rodent species make the findings more robust. Pleased to see that the codes are publicly available on GitHub. I only have a couple of minor points listed below:

Lines 367-370. I am not fully clear what the authors want to say. Please rewrite.

We have reorganized and rewritten this portion of our manuscript as follows:

“The detection parameters of our fit occupancy model implied that a single trap-night within a home had a 38% chance of detecting *M. natalensis*, conditional on the species being present (99% HPDI = [0.11, 0.71]). Further, our results indicated that increased trapping effort should generally increase *M. natalensis* detection (0.42, [-0.31, 1.17]). As such, deployment of three traps on a given night would be expected to boost mean detection over 50% (0.57, [0.44, 0.71]), while seven traps would be required to achieve a mean detection probability of at least 80% (0.83, [0.30, 1.00]), again conditional on *M. natalensis* being present. As expected, *M. natalensis* detection decreased as progressively more *M. natalensis* were captured and removed from a given house (-1.10, [-1.75, -0.45]).”

Figure S3 and S5. The season effect parameter change direction when they use house trapping data and all trapping data as discussed in line 341-343. I am wondering if this reflect the fact that the detection of *Mastomys N.* inside houses changes with season and in opposite directions compared to proximity field (see E. Fichet-Calvet et al., “Fluctuation of abundance and Lassa virus prevalence in *Mastomys natalensis* in Guinea, West Africa,” *Vector Borne Zoonotic Dis.*, vol. 7, no. 2, pp. 119–28, Jan. 2007, doi: 10.1089/vbz.2006.0520.). I am aware that in S5 the author used all trapping data (and not outside house data), but maybe the author could discuss this.

Thank you for this careful observation. To address this comment, we first note that in this revised version of the manuscript, a couple of minor analytical changes have slightly altered our seasonality results. First, in response to comments from Reviewer #3, we have redefined our seasonality variable such that three site visits (out of 72 total) have been recategorized from dry season visits to rainy season visits. In addition, while reconsidering our analyses during this revision, we realized it would be appropriate to include visit-level varying intercepts in our visit-level models (the visit-level models themselves were completely new to our previous round of revision where we needed to generate them in order to incorporate seasonality into our models).

With these tweaks to our analyses, the rainy season effect for the house trapping dataset is -0.19 [-0.68, 0.31]. The rainy season effect for the all trapping dataset is 0.16 [-0.50, 0.82]. Thus, as the Reviewer points out, the effects are in opposite directions, but crucially, both are relatively small and uncertain (they overlap substantially with 0). Further, in our house-level models, which contain only house trapping data from Sierra Leone, the rainy

season effects are overlapping 0 but in the positive direction, which contrasts with our visit-level house trapping model that includes house traps from both Sierra Leone and Guinea. Thus, our takeaway from these results is that the rainy season effects in our models are generally weak and potentially sensitive to the particular datasets examined. In contrast, the *R. rattus* effect that is the focus of our paper is stronger and consistently negative across all analyses. As such, we prefer to avoid speculating at length on the seasonality effects arising from our analyses.

Accordingly, we have revised a paragraph in our Discussion to read:

“In our models, we accounted for potential seasonal effects on *M. natalensis* captures, given that prior research has emphasized environmental influences on *M. natalensis* trap success and Lassa virus dynamics [5, 28, 45]. However, in our analyses, seasonality effects were relatively small, uncertain (all overlapped with 0 in the 99% HPDI), and sensitive to the particular dataset used for modeling. By contrast, the mean *R. rattus* presence effect was negative and larger in magnitude than the seasonality effect in all models we fit. Therefore, our results suggest that *R. rattus* presence may be more important in determining *M. natalensis* distribution and abundance than well-established environmental factors like seasonality.”

Figure S7 and S8 seems to have the same caption (both “only house trapping”), I presume one of them used all trapping data.

Thank you for drawing our attention to this potential confusion. These two figures correspond to our two house-level models: one with a site-level *R. rattus* predictor and one with a house-level *R. rattus* predictor. We have now clarified in the figure captions that these trace plots correspond to two different house-level models with different predictors.

Reviewer #2 (Remarks to the Author):

Dear Authors

Thank you for taking comments positively and changing the manuscript constructively. I am happy that all my concerns have been adequately addressed. I will revise my view now to publish.

Reviewer #2 (Remarks on code availability):

I have checked a sample and they look fine.

We greatly appreciate the Reviewer's time and energy in providing comments that helped us to significantly improve our manuscript.

Reviewer #3 (Remarks to the Author):

General Comment

The authors made substantial efforts to address the concerns raised by the reviewers regarding the first version of the manuscript. However, I still have several concerns regarding the content of the manuscript which are detailed below.

-Caution regarding the scope of the conclusions

I would recommend that the authors be extremely cautious with the formulation of the results regarding the reduction of LASV spill-over risk, especially as they could result in supporting worrying approaches of biocontrol that create more risks than they eliminate. Also - and although the authors do comment on the elements available to support the absence of role of *R. rattus* as a potential vector for LASV in the manuscript- it is challenging to assess the real risk of *R. rattus*-mediated LASV transmission as (1) very little field and experimental data is available and (2) the ability of LASV to adapt to this host - currently considered a dead end - is unknown. In light of these considerations I would recommend formulating this manuscript much more cautiously with regard to the scope of their results and adding a specific paragraph to detail the limitations of the study's conclusions in the discussion.

In addition to a paragraph dedicated to caveats regarding our overall study findings (paragraph beginning: “We also highlight several caveats that may complicate the relationship we suggest between *R. rattus* and Lassa virus spillover risk...”), we now include the following explicit caveat about biocontrol in our Discussion section:

“Nonetheless, given the potential for *R. rattus* to harm human and ecosystem health in other ways, we stress that we do not advocate for biocontrol strategies that would attempt to use this species to manage Lassa virus spillover. For example, it is critical to recognize that our findings of reduced spillover at sites with *R. rattus* are specific to the Lassa fever system and that invasive *Rattus* do host other zoonotic pathogens that can threaten humans [16, 53]. Further, the negative impacts of invasive *R. rattus* on native ecosystems, for example through predation and competition, are widespread and well-documented [48, 54].”

-Effect of seasonality

In the new version of the manuscript, the authors account for seasonality in their models by adding a specific binary predictor relative to wet season sampling, which only partly addressed my concerns on potential missing variables. The criteria used to discriminate between dry and wet seasons are not defined explicitly in the text which makes it difficult to assess the relevance of the new set of variables, this needs to be defined somewhere in the methods. Also, as with climate change, abnormal seasonal temperatures and precipitation are increasingly observed, I would recommend directly including environmental variables (temperature, precipitation, etc) that underlie the seasonal effect previously observed on *M. natalensis* populations to make the

model more explicit and remove potential biases due to e.g., a wet season that would be abnormally dry and/or cold.

Previously, our categorization of site visits into rainy or dry season sampling followed existing literature (Demby et al. 2001, *Vector Borne and Zoonotic Diseases*) that described the rainy season in Guinea as May through October. Thus, we previously coded any visits occurring May through October as rainy season sampling and any visits occurring November through April as dry season sampling.

However, the Reviewer's helpful comment led us to further investigate our binary categorization of seasonality. First, we used publicly-available datasets to pull monthly precipitation values (CHIRPS dataset) and mean daytime temperature (MODIS dataset) for the locations and times of the site visits in our dataset. These data are now included as a new supplementary Figure S13. Examination of the environmental data from our site visits led us to recognize that November site visits should be considered as having occurred in the rainy season. This led to the recategorization of sampling season for 3 of the 72 site visits in our data (3 November visits changed from dry season to rainy season). As Figure S13 illustrates, with this new rainy/dry season categorization, the rainy/dry binary variable perfectly separates site visits based on monthly precipitation (site visits occurring in our rainy season always experienced more monthly rainfall than site visits occurring in our dry season). Further, rainy season site visits had significantly lower daytime temperatures than site visits from the dry season. In sum, we are confident that our new rainy/dry season variable accurately categorizes site visits into the appropriate season and that this variable captures important variation in environmental conditions.

We have re-run all analyses in the paper using this newly-defined seasonality variable. Importantly, models again show that sampling season is less influential on *M. natalensis* catch per trap than the presence of *R. rattus*.

To clarify for our readers, we have added the following text at line 362:

“While our primary aim was to understand the effect of *R. rattus* on *M. natalensis* captures, we also wanted to account for potential seasonal effects on *M. natalensis* abundance and activity that could influence capture success [28,45]. As such, we first organized the trapping data at the visit-level, with each observation representing trapping results from a given site visit (n = 71 unique site visits that involved house trapping) and each visit coded as having occurred in the dry season (December–April) or rainy season (May–November). Our dry/rainy season definition closely aligns with prior literature [28,45,57], and we additionally confirmed using remotely-sensed data that precipitation [66] and temperature [67] differed significantly between dry and rainy season site visits using this definition of seasonality (Fig. S13).”

-Specificity of the effect

The authors controlled for the specificity of the effect of *Rattus rattus* on *Mastomys natalensis* populations at the site level by including other common rodent species as alternative predictors

of *M. natalensis* captures. This correctly addressed my concerns regarding the site-level analysis; however, it would also be necessary to confirm the specificity of the effect of *R. rattus* on *M. natalensis* for the house-level and spill-over risk analyses.

We have now added supplemental analyses to confirm the specificity of the *M. natalensis* effect for our two house-level models (Figures S6 & S8) as well as for our spillover risk analysis (Figure S10). In all cases, the mean value of the *R. rattus* effect is larger in magnitude than any other rodent species tested, and in all cases the posterior distributions for all other rodent effects overlap 0 in the 80% HPDI. Thus, we conclude there is no evidence that local rodent species other than *R. rattus* have consistent negative effects on catch per trap of *M. natalensis*.

REVIEWERS' COMMENTS

Reviewer #3 (Remarks to the Author):

The authors have -again- made substantial efforts to address my main concerns, and improved their manuscript. It is now suitable for publication.

REVIEWER COMMENTS

Reviewer #3 (Remarks to the Author):

The authors have -again- made substantial efforts to address my main concerns, and improved their manuscript. It is now suitable for publication.

We are grateful to the Reviewer for their attention and effort in helping us to craft a stronger manuscript, and we are pleased to see they now judge our manuscript worthy of publication.